# A cost comparison of various hourly-reliable and net-zero hydrogen production pathways in the United States

Justin M. Bracci[1,3], Evan D. Sherwin [1,4], Naomi L. Boness[2] & Adam R. Brandt [1] ✉

Hydrogen ($H_2$) as an energy carrier may play a role in various hard-to-abate subsectors, but to maximize emission reductions, supplied hydrogen must be reliable, low-emission, and low-cost. Here, we build a model that enables direct comparison of the cost of producing net-zero, hourly-reliable hydrogen from various pathways. To reach net-zero targets, we assume upstream and residual facility emissions are mitigated using negative emission technologies. For the United States (California, Texas, and New York), model results indicate next-decade hybrid electricity-based solutions are lower cost ($2.02-$2.88/kg) than fossil-based pathways with natural gas leakage greater than 4% ($2.73-$5.94/kg). These results also apply to regions outside of the U.S. with a similar climate and electric grid. However, when omitting the net-zero emission constraint and considering the U.S. regulatory environment, electricity-based production only achieves cost-competitiveness with fossil-based pathways if embodied emissions of electricity inputs are not counted under U.S. Tax Code Section 45V guidance.

To mitigate global greenhouse gas (GHG) emissions and halt global temperature rise, net-zero emissions technology must be scaled rapidly. While electrification coupled with increased renewable energy deployment can aid in eliminating emissions in buildings, passenger vehicles, and other low-temperature heating applications, other zero-emission fuels may be needed for hard-to-abate sectors such as the cement and steel industries, along with ocean shipping and long-distance ground transportation[1–4]. Hydrogen ($H_2$) is one such zero-emission fuel that could be used to mitigate emissions in these sectors as well as others. The International Energy Agency (IEA) estimates that global hydrogen demand could increase from 94 million metric tons (MMT) in 2021 to 660 MMT by 2050 and could generate cumulative emissions reductions of 80 billion metric tons (BMT) carbon dioxide ($CO_2$) by 2050[4,5].

To achieve this level of emissions reductions, hydrogen must be produced in a near-zero emission and reliable manner. This will be a challenge, as 98% of current global hydrogen production is from fossil-based pathways without installed $CO_2$ capture[6], and developing

electrolysis pathways that use intermittent renewable electricity are unreliable without incurring costs for daily and seasonal storage. In many prior hydrogen studies, reliability distinctions are unclear or not addressed[2,7–10], and studies that do address reliability tend to have coarse methods of joining techno-economic analysis and life-cycle assessment[11–13].

In this study, we conduct a consistent intercomparison of the cost of producing reliable, net-zero hydrogen under various electricity-based and fossil-based production pathways. First, we include a $CO_2$ equivalent ($CO_2e$) removal cost that ensures both fossil-based and electricity-based hydrogen production pathways have net-zero life-cycle emissions, thus making a fair comparison between different technologies. This avoids a situation where hydrogen production pathways with different GHG intensities are compared on a simple cost per unit of hydrogen, without accounting for the differences in associated emissions. The zero-emission constraint is also reflective of national policies and corporate commitments made around the world, therefore making our findings decision-relevant. Second, our model

[1]Department of Energy Science & Engineering, Stanford University, Stanford, CA, USA. [2]Precourt Institute for Energy, Stanford University, Stanford, CA, USA. [3]Present address: National Renewable Energy Laboratory, Golden, CO, USA. [4]Present address: Lawrence Berkeley National Laboratory, Berkeley, CA, USA. ✉e-mail: abrandt@stanford.edu

accounts for the consistency of hydrogen production as a function of time. This is a major factor for electrolytic hydrogen pathways, which generally aim to leverage intermittent low-cost renewable electricity resulting in a hydrogen production profile that can fluctuate on an hourly, daily, and seasonal basis. Using such sources could be a challenge for end consumers requiring consistent delivery. By explicitly addressing inconsistencies between hydrogen production pathway emissions and delivery reliability, this work can help to better inform hydrogen infrastructure investment decisions.

## Results

### Methods overview
We start by reviewing possible electricity-based and fossil-based hydrogen production systems. For each of these production pathways, we conduct a life-cycle GHG emission assessment and techno-economic assessment assuming next decade technology and emissions projections for facilities located in California (Sacramento), New York (Albany), and Texas (El Paso). These locations were chosen based on their differing geographic and energy attributes. We set the production facility size to 250 metric tons per day of hydrogen (roughly equal to 500 MW$_e$ electrolysis at full capacity), a typical size of hydrogen production plants at petroleum refineries[14], to reflect a next-decade future with growing hydrogen demand and economies of scale benefits. For the electricity-based hydrogen production pathways, a least-cost linear optimization model was developed and utilized for both the emissions assessment and techno-economic modeling. The model is hourly resolved with a total of 8760 timesteps representing each hour of a specified year. All electricity-based production pathways explored in this study consider an onsite-solar photovoltaic (PV) facility with the option to include energy storage (battery or compressed hydrogen storage at 200 bar [20 MPa]) and/or a grid interconnection to increase delivery reliability. Hourly grid electricity prices are derived from the National Renewable Energy Laboratory's (NREL) Mid-Case scenario in the Cambium database[15] and hourly solar capacity factor data is derived from NREL's System Advisor Model[16]. Water use is also considered, although, we do not go as far as to consider how this cost changes by location given geographic variability in water access and water quality. GHG emissions from electricity input are accounted for on a life-cycle basis; meaning embodied emissions (emissions from materials extraction, manufacturing, construction, etc.) for solar PV and other grid electricity sources are considered in the electricity-based pathways. See Fig. 1 for further detail on the system configuration as well as emissions considerations.

For the fossil-based production pathways, a simulation-based model was developed and utilized for the GHG emission and techno-economic assessments. Fossil-based hydrogen production pathways explored include steam methane reforming (SMR), steam methane reforming with carbon capture and storage (SMR-CCS), and autothermal reforming with carbon capture and storage (ATR-CCS). GHG emissions accounted for in the fossil-based pathways include those directly emitted from the production plant as well as life-cycle emissions (including embodied emissions) of the input electricity and natural gas. See Fig. 2 for more detail on the fossil-based hydrogen production system configuration and GHG emission inclusions.

To make each production pathway net-zero emissions, the facility operator incurs a $CO_2$ removal cost for any residual emissions after $CO_2$ capture is performed to finance corresponding negative emissions technology, as well as to mitigate any energy feedstock-derived emissions. This cost is included in the techno-economic model and is considered a cost of production. All scenarios assume a next-decade emission removal cost of $200 per metric ton $CO_2$e[17–20], representing a technology agnostic cost as the details of offsite removal are beyond the scope of this paper (e.g., some method to mitigate $CO_2$ equivalent emissions, such as air capture, soil carbon amendment, reforestation, mineralization,

etc.). The value of $200 per metric ton $CO_2$e is the same as what is used in the next-decade scenarios from Sherwin et al. 2021[19].

The level of reliability is also kept consistent between hydrogen production pathways to make fair LCOH comparisons. For this analysis, we assume each production pathway must reach hourly reliability. We also assume that all fossil-based pathways are already hourly-reliable. This assumption is valid since many fossil-based hydrogen production facilities today are located at refineries and ammonia production facilities, and operate at or near 100% capacity factor with downtime for scheduled maintenance staggered to facilitate continuous hydrogen deliverability[21,22].

Lastly, we consider the impacts of the U.S. Inflation Reduction Act (IRA)[23] on the LCOH for each hourly-reliable hydrogen production pathway explored. In this section of our analysis, we remove the zero-emission constraint, thus eliminating the cost for removing all life-cycle emissions, and include tax credits available under Sections 45V and 45Q of the IRA. This enables a cost comparison between viable hydrogen production pathways given a regulatory credit market rather than a binding emissions constraint. More detail on the 45V and 45Q incentive schemes is provided in the results section.

See Methods section and supporting information (SI) for a detailed discussion of modeling methods and input assumptions[24]. The Methods section lists each of the primary input values used in the electricity- and fossil-based cost models. All other input values are provided in the SI and source data[24,25].

### Reliable electricity-based hydrogen production cost analysis
In many electricity-based hydrogen cost modeling efforts, the reliability of hydrogen delivery is either not considered or is unclear. This typically implies that cost results are only presented at yearly levels of reliability, interpretable as the cost of producing some total amount of hydrogen over a year[2,7–10]. That is, in such cases a facility is modeled to produce output over a given year, but if PV electricity is used, actual output will vary over the course of the year (more output in daytime and summer, less at night and winter). Such output variability can be mitigated by storing electricity as input to hydrogen generation or by storing the generated hydrogen.

To avoid this effect, we model the same size facility (with average annual output of 250 metric tons of hydrogen per day) with differing reliability constraints. In the yearly case, the facility is required to supply 91,250 metric tons per year of hydrogen. In the monthly case, it must supply 91,250/12 or 7604 metric tons each month, while in the daily reliability case, it must supply 91,250/365 or 250 metric tons each day, and so on. Figure 3 shows the next-decade LCOH produced via electrolysis of water, coupled with a co-located solar PV and on-site battery and hydrogen storage to allow the facility to meet various levels of reliability at three different locations in the United States. From the pathways with dedicated PV solar and no grid connection, Yearly reliable hydrogen delivery is lowest cost ($2.88–$3.88/kg $H_2$), with total LCOH increasing by 106%, 97%, and 35% in the Hourly case for California, New York, and Texas, respectively. Noticeably, total LCOH for the Hourly case in Texas increases the least due to the facility being located at a lower latitude with less variation in solar PV output throughout the year. This finding would also apply to locations outside of the United States that are located at lower latitudes.

Figure 3 shows that yearly reliable hydrogen delivery has the lowest cost because hydrogen and electricity storage are not required. Hydrogen produced during the day using solar PV electricity will be delivered to the consumer with no regard to when the consumer will use the hydrogen. To better meet consumer needs, monthly- or daily-levels of delivery reliability is needed and requires an overbuild of the solar PV and hydrogen storage components to ensure extra energy produced in the summer from the solar PV can be shifted to the winter. If hourly-reliable hydrogen delivery is necessary, a more dramatic overbuild of both the solar PV and energy storage components is

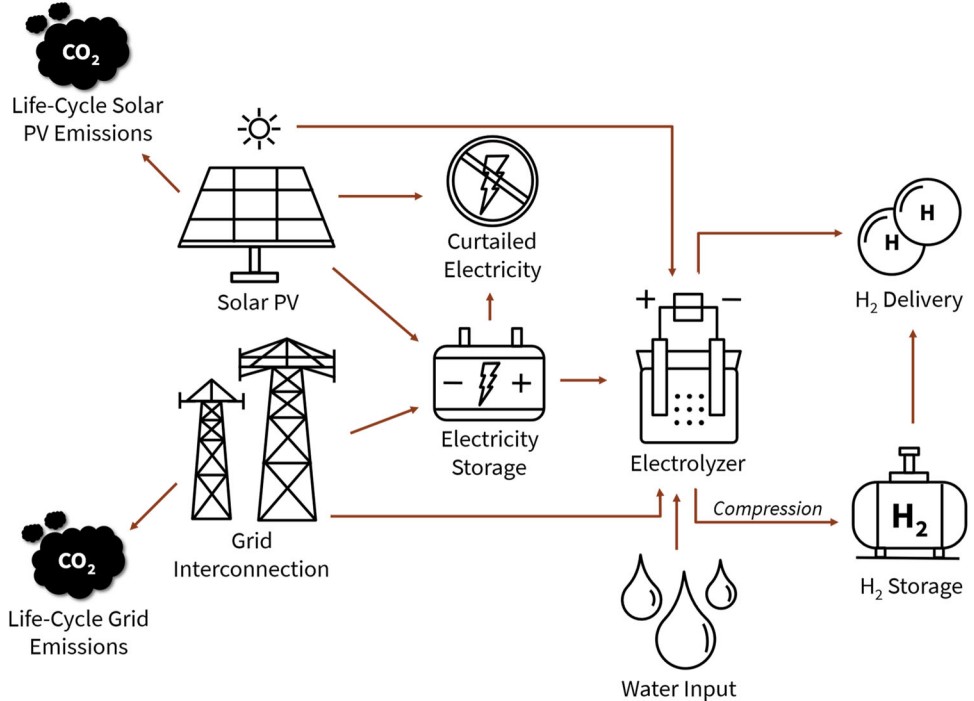

**Fig. 1 | Electricity-based hydrogen production system configuration diagram.** PV refers to photovoltaic.

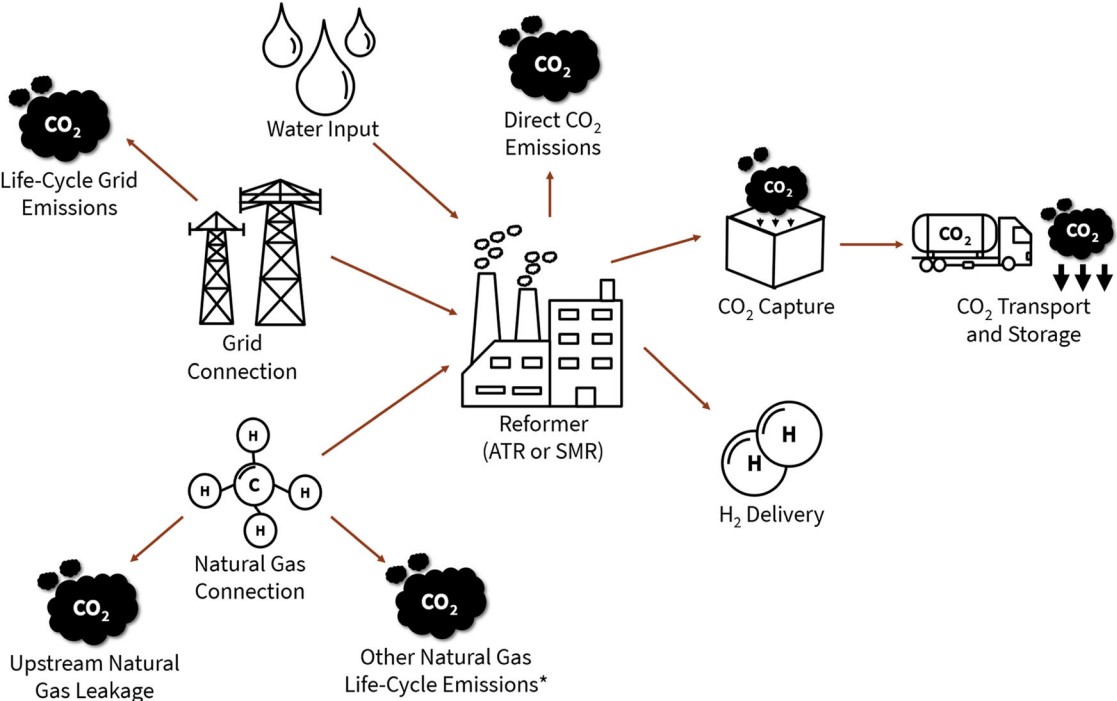

**Fig. 2 | Fossil-based hydrogen production system configuration diagram.** *Refers to combustion emissions related to natural gas production and processing as well as any other natural gas embodied emissions. ATR refers to auto-thermal reforming. SMR refers to steam methane reforming.

required to ensure both seasonal and daily energy shifting. For example, in the Yearly case for California, only 2.1% of electricity produced from the solar PV facility goes unused by the electrolyzer, but as reliability constraints shift to monthly, daily, or hourly levels, the percent of solar PV electricity that is curtailed increases to 31%, 38%, and 39%, respectively. In terms of storage requirements, the Yearly case requires no storage, while the Monthly, Daily, and Hourly pathways require 2028 metric ton, 2332 metric ton, and 2502 metric ton of

$H_2$ storage, respectively. Battery electric storage installment is minimal for each reliability case due to the lower relative cost of storage using hydrogen tanks (see Methods section below for cost data). The $CO_2e$ removal cost tends to increase as we shift from yearly to hourly reliability because embodied solar PV emissions increase as the solar PV facility is overbuilt. Refer to Table 1 for all electricity-based pathway system component sizing results for the Sacramento, California location. Component sizing results for the other locations analyzed are

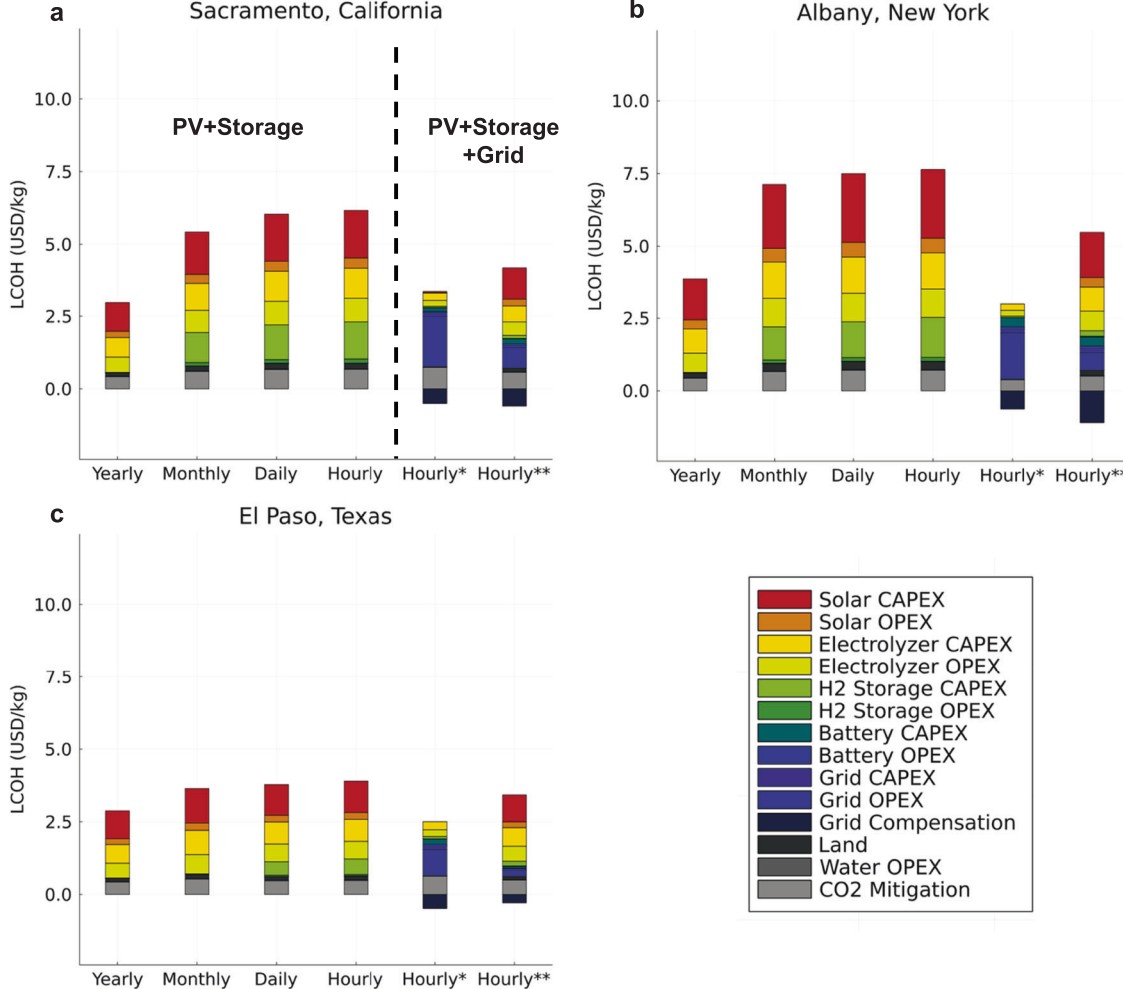

**Fig. 3 | Impact of hydrogen reliability on LCOH.** Levelized cost of net-zero hydrogen produced from electricity-based pathways using electricity from a solar PV facility coupled with energy storage under various levels of reliability and for facilities located in (**a**) Sacramento, CA, (**b**) Albany, NY, and (**c**) El Paso, TX. *Refers to an hourly reliable pathway that uses solar PV, energy storage (hydrogen and battery storage), with an unconstrained grid connection. **Refers to an hourly reliable pathway that uses solar PV, energy storage, and a maximum of 10% electricity from the grid. LCOH refers to the levelized cost of hydrogen production in units of United States dollars per kg (USD/kg). PV refers to photovoltaic. CAPEX refers to capital expenditures. OPEX refers to operation expenditures.

available in Section 9 of the SI with trends remaining consistent for each of the three locations analyzed[24].

The two rightmost stacked bar charts in each subplot of Fig. 3 represent hourly reliability electricity-based production pathways that also have an accessible electric grid connection. The Hourly* case has unlimited access to the electric grid (PV/Storage/Grid*) while the Hourly** case is constrained to 10% electricity usage from the electric grid for hydrogen production operations (PV/Storage/Grid**). As

shown, an unconstrained electric grid connection allows for hourly delivery reliability at a 48-69% lower cost compared to the Hourly pathway, depending on the state. We also notice in the Hourly* case that electricity input for production comes almost exclusively from the grid due to near-zero marginal pricing during hours when excess renewables would otherwise be curtailed[15]. While not as dramatic, the Hourly** pathway also results in cost savings compared to the off-grid Hourly case, with an LCOH reduction of 42% in California, 19% in Texas, and 43% in New York. These cost savings are realized because an overbuild of solar PV and hydrogen storage is no longer necessary and excess on-site electricity can be sold back to the electric grid when prices are high instead of being curtailed.

To support this finding, we again examine operations and component sizing for the California location (see Table 1). In this location, we find only 92 metric tons and 211 metric tons of hydrogen storage are used in the Hourly* and Hourly** pathways, respectively. These are an order of magnitude lower than the storage in the off-grid Hourly case (2502 metric tons). However, these two grid-tied pathways also take advantage of small battery storage systems, equivalent to 11 metric tons hydrogen storage (584 MWh$_e$ battery storage) in the Hourly* pathway and 13 metric tons (684 MWh$_e$ battery storage) in the Hourly** pathway, to store excess solar PV electricity and low-price grid electricity for later use. The average price of electricity sold back to the

**Table 1 | Electricity-based hydrogen production pathway system component sizing for a facility located in Sacramento, California**

| Component | Yearly | Monthly | Daily | Hourly | Hourly* | Hourly** |
|---|---|---|---|---|---|---|
| Battery Storage (MWh$_e$) | 0 | 0 | 0 | 0 | 584 | 684 |
| Hydrogen Storage (metric ton H$_2$) | 0 | 2028 | 2332 | 2502 | 92 | 211 |
| Solar PV (MW$_e$) | 1979 | 2842 | 3158 | 3195 | 94 | 2103 |
| Electrolyzer (MW$_e$) | 1554 | 2200 | 2408 | 2406 | 590 | 1337 |
| Grid Connection (MW$_e$) | 0 | 0 | 0 | 0 | 590 | 394 |

grid in the Hourly* and Hourly** pathways amounts to $1130/MWh of electricity (MWh$_e$) and $66/MWh$_e$, respectively, while the average price of electricity bought from the grid in these cases equals $19/MWh$_e$ and $24/MWh$_e$, respectively. These price differences show how important the interaction between the grid and battery storage system can be for producing low-cost, electricity-based hydrogen. Again, these trends remain consistent for the Texas and New York production locations. Explore Section 9 of the SI for more granular detail on electricity-based hydrogen production operations[24].

It is also important to highlight that there are costs incurred to connect to and use the grid system, as well as to remove grid-related emissions, in the Hourly* and Hourly** pathways that are not found in the off-grid Hourly pathway. However, the costs associated with using grid electricity are less than the savings created using that same electricity. This makes the grid connection beneficial for reducing the total LCOH, while also maintaining hourly delivery reliability. Although, the magnitude of cost savings for the grid-tied pathways in Fig. 3 will

depend on the evolution of hourly electricity prices as the grid is decarbonized and likely will not be precisely the same as the price forecasts used in this study. If the grid hourly prices become more extreme in off-peak, non-solar-generating hours (a plausible assumption), then some of the savings associated with grid interconnections may not arise.

See Figure S.23 of the SI to examine how the next-decade results shown in Fig. 3 compare to those for current and mid-century timeframes[24]. LCOH trends seen in Fig. 3 remain consistent for current and mid-century results but with differing magnitudes due to the level of technology evolution.

**Net-zero, hourly-reliable hydrogen production cost comparison**
Figure 4 compares the electricity-based and fossil-based LCOH results with the following assumptions: next-decade technology costs, location-specific energy attributes, net-zero emissions, and hourly delivery reliability. The LCOH is composed of three parts: the cost of

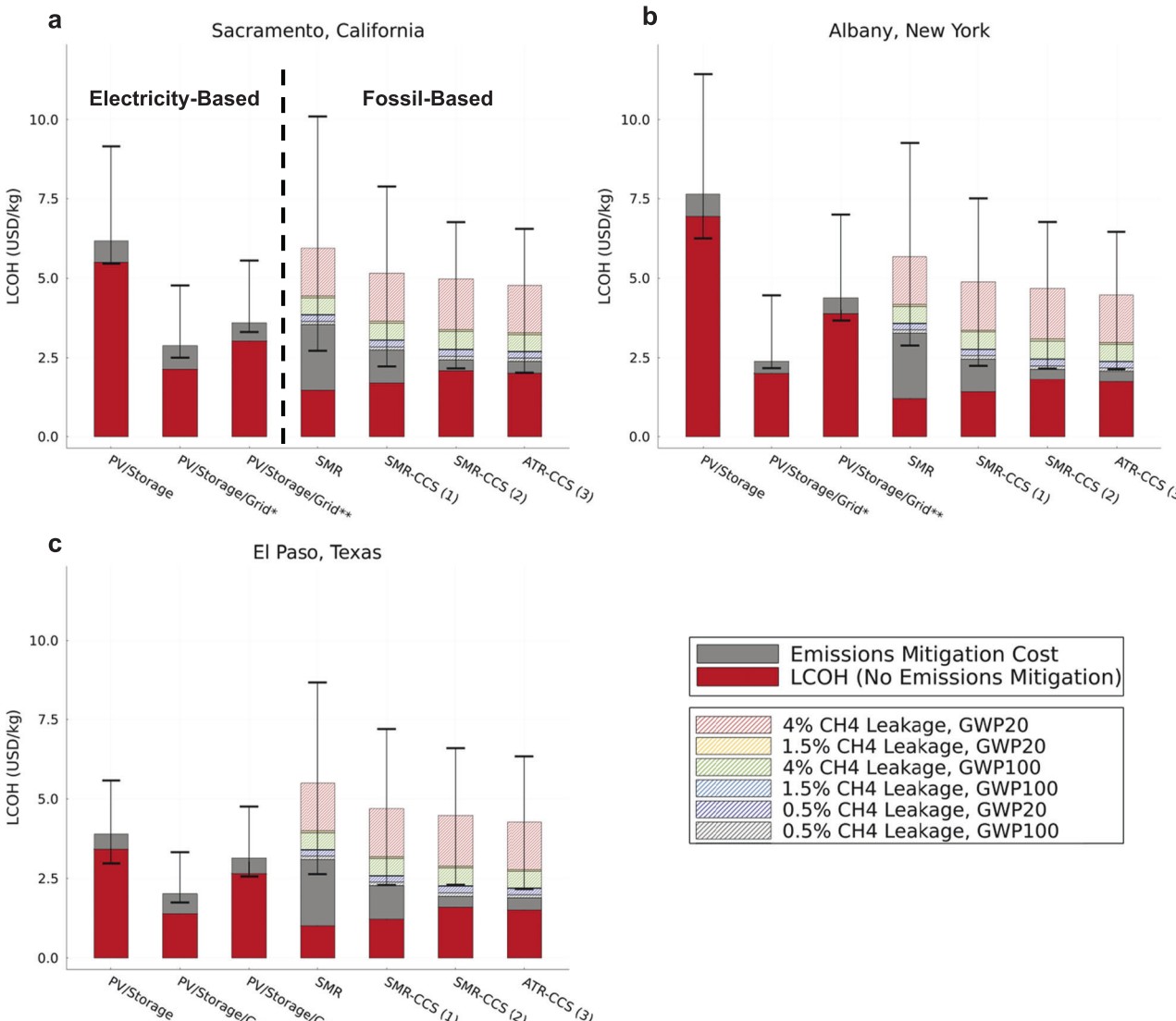

**Fig. 4 | Comparing hourly-reliable, net-zero hydrogen production costs.** Hourly-reliable, net-zero hydrogen production cost comparison between electricity-based and fossil-based production pathways assuming next decade technologies and facility locations in (**a**) Sacramento, CA, (**b**) Albany, NY, and (**c**) El Paso, TX. *Refers to an electricity-based hydrogen production scenario with possible on-site solar PV, energy storage, and an unconstrained grid connection. **Refers to an electricity-based production scenario with on-site solar PV, energy storage, and constrained grid use. (1) SMR-CCS with process $CO_2$ capture, (2) SMR-CCS with process and flue gas $CO_2$ capture, (3) ATR-CCS with process $CO_2$ capture. LCOH refers to the levelized cost of hydrogen production in units of United States dollars per kg (USD/kg). PV refers to photovoltaic. GWP refers to global warming potential. SMR refers to steam methane reforming. CCS refers to carbon capture and storage. ATR refers to auto-thermal reforming.

the hydrogen production facility and operations, the cost of removing life-cycle feedstock and direct facility emissions (excluding emissions from natural gas leakage), and the cost of removing upstream natural gas leakage. These last two costs are treated as separate categories to provide insights into the importance of natural gas leakage on hydrogen production emissions mitigation. With these considerations, Fig. 4 shows that electricity-based hydrogen production that uses a combination of energy storage, solar PV, and grid electricity can be at cost-parity, if not lower cost, than all fossil-based alternatives when accounting for emission removal costs. This takeaway is true regardless of location and, in Texas, the PV/Storage/Grid* (same as Hourly*) case would even be viable if there was no natural gas leakage in the fossil-based cases. The fossil-based production pathways become especially uneconomic from a net-zero perspective when upstream natural gas leakage exceeds 4%. In our baseline fossil-based pathways, $CO_2$ removal costs for mitigating 4% natural gas leakage can contribute up to $2.55/kg to the net-zero emission LCOH, more than doubling the LCOH when no cost for removing residual emissions is included[22]. This assumes a 20-year global warming potential (GWP) timeframe is used for natural gas, but the $CO_2$ removal cost for 4% natural gas leakage is still relatively high ($0.84/kg–$0.90/kg) using a 100-year GWP timeframe (see Fig. 4). This point on the emission intensity of natural gas leakage has been highlighted in other studies as well[26,27]. However, if cost penalties for emissions are left out, fossil-based pathways remain lower-cost or near cost-parity with electricity-based alternatives considered in this study.

The error bars shown in Fig. 4 are 96% confidence intervals constructed using Monte Carlo simulation. We find from the error bars that the inclusion of a grid connection to the electricity-based production pathways reduces cost uncertainty and enables higher cost certainty than fossil-based pathways. Section 7 of the SI gives more detail on the error bar construction[24].

Figure 4 also shows that the SMR-CCS (2) and ATR-CCS pathways, with a high percent carbon capture (96% and 95% capture, respectively), have a lower total LCOH than the SMR and SMR-CCS (1) pathways that have no or little carbon capture equipment installed (0% and 56% capture, respectively). This can be attributed to the large residual emissions that must be removed when limited or no carbon capture equipment is installed at a fossil-based facility. However, when an emission removal cost is not considered, and fossil-based pathways are not zero-emission, the pathways with limited or no carbon capture still have the lowest cost.

Figure S.24 of the SI also contains Fig. 4 but is accompanied by cost comparisons using current and mid-century technology data[24]. The current technology results in Figure S.24 show a similar trend to those in Fig. 4 with the PV/Storage/Grid* pathway being favorable over fossil-based pathways when natural gas leakage is reaching 4%. It is also noticeable how uneconomic hourly-reliable hydrogen production costs are in the current timeframe when we set production to net-zero emissions. In contrast, the mid-century results show little additional cost for emissions removal since technology development itself will push hydrogen production pathways closer to net-zero. Note that there is inherent uncertainty in future energy system costs and so results in Figure S.24 should not be taken as forecasts, but as plausible scenarios defined using existing literature.

## Model input parameter sensitivity analysis

Exploring LCOH sensitivities to each of the primary input parameters of the model, tornado diagrams are shown in Fig. 5. Figure 5a displays the parameter sensitivity results for the PV/Storage/Grid* (Hourly*) hydrogen production pathway from Fig. 4. We run the electricity-based hydrogen cost model for this production pathway with realistic low- and high-end input parameter values that are generally representative of current and projected mid-century technology, respectively. As shown, uncertainty in the grid demand charge can cause the

most dramatic fluctuation in LCOH with a possible LCOH change of $1.67–1.90/kg depending on the location. This finding points to potential advantages of developing grid-connected hydrogen production plants in locations with low or no demand charges. Demand charges are also projected to decrease over time due to continued cost reductions in battery storage technologies which should make it easier to find such locations[28]. To capture uncertainty and variability in hourly grid electricity prices, we use NREL's 2035 Cambium Mid-Case as our central scenario and the 2035 Low Renewable Energy Cost and High Renewable Energy Cost scenarios as the low and high price bounds, respectively[15]. We find using these electricity price scenarios does not result in as large of a change in LCOH as that of the grid demand charge. However, on a relative basis, the LCOH is more sensitive to average electricity price than it is to grid demand charge. This finding is shown in Figures S.1 of the SI which contains spider plots for the PV/Storage/Grid* production pathway[24]. Figure S.2 contains similar spider plots, but for the ATR-CCS production pathway. These spider plots show the change in LCOH due to the relative change (% change) in input parameter values, providing insight into the rate of cost sensitivity for each parameter. See the Methods section for more detail on the electricity price scenarios used in this study.

$CO_2$ removal cost uncertainty can also impact the LCOH in the PV/Storage/Grid* pathway ($0.92–$1.88/kg variability depending on the state). This indicates that continued development of negative emission technology (e.g., direct air capture, bioenergy with carbon capture and storage, other nature-based solutions, etc.) is imperative to enable low-cost, net-zero, electricity-based hydrogen production. Continued solar and electrolyzer technology advancements (e.g., suitable rare earth material replacements), and cost reductions, (e.g., production process streamlining), are also important for minimizing electricity-based hydrogen production costs. However, in this specific production pathway, grid electricity use dominates over solar PV-derived electricity, meaning that changes in solar PV cost results in minor changes in LCOH. In addition, we notice that uncertainty in hydrogen storage capital cost does not dramatically impact the PV/Storage/Grid* pathway cost. This can be attributed to additional reliability enabled by a grid connection and means that it is not imperative to have low-cost hydrogen storage to keep hydrogen production cost low in an electricity price and emissions intensity environment similar to the one modeled here.

Figure 5b displays a tornado diagram for input parameters from the ATR-CCS hydrogen production scenario, assuming a baseline case of 1.5% natural gas leakage under a 20-year global warming potential (GWP) estimate (85 kg $CO_2$e per kg $CH_4$)[29]. We choose the 20-year GWP timeframe as the baseline, as opposed to the 100-year GWP timeframe (30 kg $CO_2$e per kg $CH_4$)[29], because of the urgency to decarbonize in the next few decades and the short-lived nature of natural gas in the atmosphere[26,30]. The impact of using a 100-year GWP timeframe is still shown in Fig. 5b and is one of the more sensitive inputs. Uncertainty in the $CO_2$ removal cost causes the most prominent fluctuation in the LCOH for ATR-CCS production, with a variability of $3.10–$3.18/kg depending on the state. This again exemplifies the need for continued research and development into negative emissions technologies if low-cost, net-zero, and fossil-based hydrogen production is to be cost-effective. We also find that natural gas leakage rate plays a large role in the LCOH, especially over the 20-year GWP timeframe ($2.39/kg variability). This should motivate industry to prioritize best practices when handling natural gas and motivate researchers and companies to continue developing technology to rapidly detect, mitigate, and prevent methane emissions, yielding benefits for the environment, human health, and, in some cases, industrial profitability. Notably, we find that uncertainty in project life does not have a large impact on fossil-based hydrogen production cost ($0.13/kg variability). This means that if fossil-based hydrogen producers wish to build new SMR-CCS or ATR-CCS facilities to act as a bridge before electricity-based hydrogen

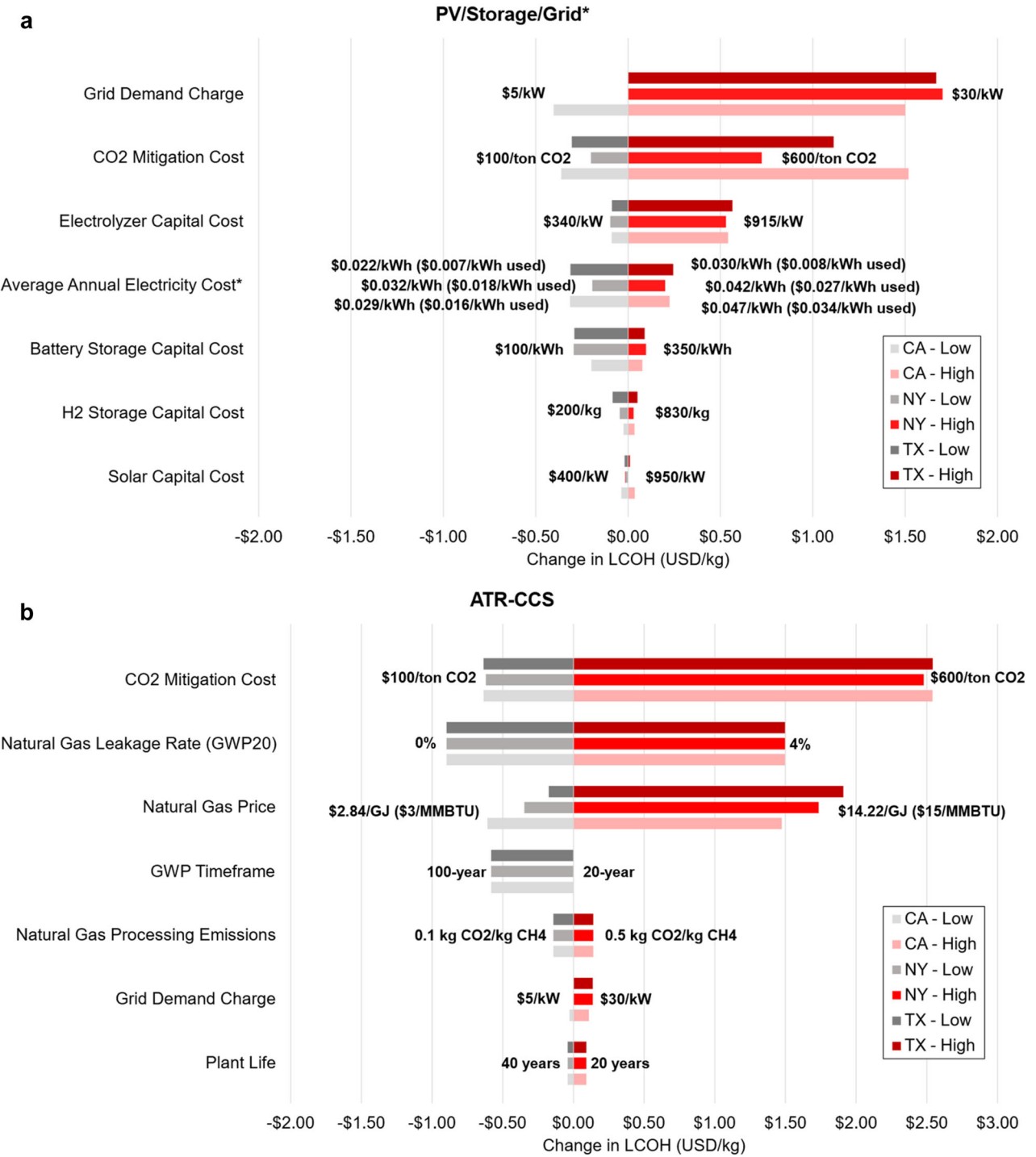

**Fig. 5 | Electricity-based and fossil-based hydrogen production sensitivity analysis. a** Sensitivity analysis for the LCOH of net-zero, hourly-reliable, electricity-based hydrogen production using solar PV, unconstrained grid electricity, and energy storage. **b** Sensitivity analysis for the LCOH of net-zero, hourly-reliable, fossil-based hydrogen production from ATR-CCS. A full table of sensitivity results can be found in Sections 5 and 6 of the SI. *Average annual electricity price and carbon intensity are different from the average annual electricity use price and carbon intensity since the optimization model will choose which hours to use grid electricity. LCOH refers to the levelized cost of hydrogen production in units of United States dollars per kg (USD/kg). GWP refers to global warming potential. PV refers to photovoltaic. CCS refers to carbon capture and storage. ATR refers to auto-thermal reforming.

production technologies mature, they can be confident that the project will be cost-effective even if the project lifetime is cut to 20 years.

Figure 5 only shows seven notable input parameter sensitivity results for PV/Storage/Grid* and ATR-CCS production pathways. Sensitivity results for other input parameters can be found in Sections 5 and 6 of the SI[24].

**Inflation reduction act impacts on hydrogen production costs**
With the passing of the Inflation Reduction Act in the United States, we separately consider the effect of 45V and 45Q tax credits. These crediting schemes operate differently compared to the above approach of applying an emission removal cost to all emissions such that the resulting hydrogen is net-zero. As stated in Section 45V of the

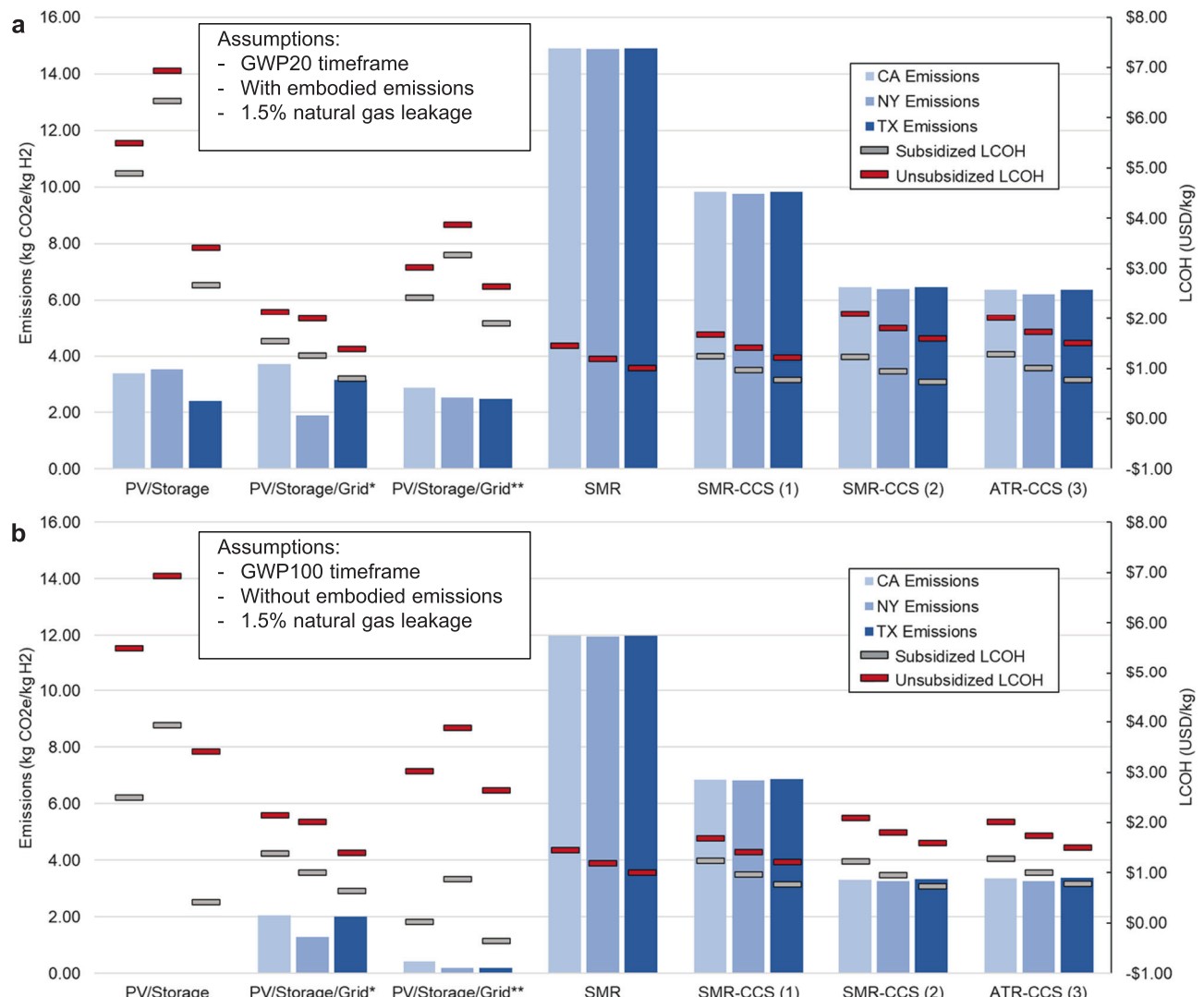

**Fig. 6 | Next-decade technology IRA emission and cost analysis. a** Total GHG emissions (in kg $CO_2$e / kg $H_2$ produced), unsubsidized LCOH (in $/kg $H_2$), and subsidized LCOH (in $/kg $H_2$) for each hourly reliable hydrogen production pathway, for each state, assuming a 20-year GWP timeframe, 1.5% natural gas leakage, and including embodied emissions of electricity-generating sources. **b** Total GHG emissions (in kg $CO_2$e/kg $H_2$ produced), unsubsidized LCOH (in $/kg $H_2$), and subsidized LCOH (in $/kg $H_2$) for each hourly reliable hydrogen production pathway, for each state, assuming a 100-year GWP timeframe, 1.5% natural gas leakage, and excluding embodied emissions. The subsidized LCOH values that consider 45V are valid for the first 10 years of project operation as defined in the IRA. The subsidized LCOH values that consider 45Q are valid for the first 12 years of project operation as defined in the IRA. LCOH refers to the levelized cost of hydrogen production in units of United States dollars per kg (USD/kg). GWP refers to global warming potential. GHG refers to greenhouse gas. IRA refers to the U.S. Inflation Reduction Act. PV refers to photovoltaic. SMR refers to steam methane reforming. CCS refers to carbon capture and storage. ATR refers to auto-thermal reforming.

Inflation Reduction Act[23], hydrogen production pathways that have life-cycle emissions less than 0.45 kg $CO_2$e/kg $H_2$ can receive $3.00/kg $H_2$ PTC for 10 years, while pathways with emissions between 0.45 and 1.5 kg $CO_2$e/kg $H_2$ can receive a $1.00/kg H2 PTC, pathways between 1.5 and 2.5 kg $CO_2$e/kg $H_2$ receive a $0.75/kg $H_2$ PTC, and pathways between 2.5 and 4 kg $CO_2$e/kg $H_2$ receive a $0.60/kg $H_2$ PTC for 10 years of operation. Fossil-based hydrogen producers have the flexibility to choose between 45V or 45Q, with 45Q providing the hydrogen producer a tax credit of $85/metric ton $CO_2$ permanently captured and sequestered for 12 years of operation. In addition, electricity-based hydrogen producers may also have flexibility to stack other tax credits available through the Inflation Reduction with 45V, but without firm guidance, we decide to omit these from this analysis[23].

Figure 6 contains the next-decade total GHG emissions (blue bars), unsubsidized LCOH (red dash), and subsidized LCOH (grey dash) for each hourly-reliable hydrogen production pathway, and for each state, under two different emissions scenarios. Figure 6a results

assume a 20-year GWP timeframe, 1.5% natural gas leakage, and the inclusion of embodied emissions of energy feedstocks while Fig. 6b results assume a 100-year GWP timeframe, 1.5% natural gas leakage, and the exclusion of embodied emissions. We vary treatment of GWP timeframe and embodied emissions in Fig. 6 as it is still unclear whether life-cycle emissions of solar PV and other electricity-generating sources will be counted and which GWP timeframe will be used when determining the level of PTC achievable under 45V. We again assume next-decade technology in Fig. 6, which in this case means the production facility is built before the start of 2033 because 45V and 45Q tax credits expire afterwards[23]. Neither the unsubsidized nor subsidized LCOH values in Fig. 6 include any costs for removing residual $CO_2$ emissions with negative emissions technologies to better reflect typical hydrogen production economics.

Looking at the electricity-based pathways in Fig. 6, the results show that a full $3.00/kg $H_2$ PTC is achievable for the hourly-reliable, PV/Storage and PV/Storage/Grid** pathways and a lower level PTC is

available for the PV/Storage/Grid* pathway, when embodied emissions are not counted. The unconstrained grid case achieves a lower-level PTC because it uses a higher percentage of electricity from the carbon intense grid. This causes the subsidized LCOH of the electricity-based pathway with a constrained grid connection to drop below that of the pathway where the grid connection is unconstrained, a much different result than what is shown in Figs. 3, 4. In contrast, when embodied emissions are counted, no hourly-reliable electricity-based pathway achieves a PTC greater than $0.75/kg and the unconstrained grid-connected pathway remains lower cost than the corresponding grid-constrained pathway. See Table S.2 in the SI for embodied emissions values of different electricity generating sources[24]. It is also notable that we did not include embodied emissions of the electrolyzer in this analysis. Including these emissions in Fig. 6a could reduce, or eliminate, the value of the PTC for electricity-based pathways further depending on the emission accounting scheme chosen for 45V.

Examining the hourly-reliable, fossil-based pathways, we find the 45V PTC is only available with the 100-year GWP assumption (Fig. 6b) and for pathways with a high carbon capture percentage (SMR-CCS (2) and ATR-CCS) because these production pathways have GHG emissions below 4 kg $CO_2$e/kg $H_2$. When comparing the 45V PTC with the 45Q tax credit for the fossil-based pathways, results show the 45Q tax credit is more lucrative for each pathway (except for the SMR pathway which receives neither 45V nor 45Q) no matter whether the 20-year or 100-year GWP timeframe is. Therefore, all subsidized LCOH results in Fig. 6 that are for fossil-based pathways are assumed to use 45Q instead of 45V. While not illustrated in Fig. 6, we also identified that natural gas leakage rate is not an important sensitivity to the subsidized LCOH of the fossil-based pathways since the 45Q tax credit is based on carbon capture rate and not carbon intensity. Given these observations, Fig. 6 shows that all fossil-based pathways with CCS are lower cost than the SMR pathway by at least $0.18/kg during the years when the projects are eligible for 45Q.

Comparing fossil-based and electricity-based subsidized production costs in Fig. 6a, where embodied emissions are included, we find that in almost every situation explored, electricity-based production remains less favorable. Only the PV/Storage/Grid* pathway comes close to cost parity with fossil-based alternatives for each state. In Fig. 6b, however, both grid-connected electricity-based solutions are at or below cost-parity with fossil-based options. In Texas, the PV/Storage option is also cost-competitive due to the favorable wind and solar resources around El Paso. These findings highlight just how important 45V emissions accounting guidance will be in determining which electricity-based hydrogen production solutions will become viable by the next decade in the United States.

Figure S.25 in the SI is the same as Fig. 6 but with current technology input parameter values[24]. The data in Figure S.25 may be more relevant to businesses looking to rapidly expand the hydrogen economy in the United States today. The raw data to generate these figures is available in the source data found in an accompanying GitHub repository[25].

## Discussion

This study highlights that net-zero electricity-based hydrogen production with variable energy input becomes increasingly expensive with stricter reliability constraints and no grid connection. Including a net-zero emission constraint to the LCOH places climate solutions with different emissions profiles on a level techno-economic playing field for comparison. Therefore, the large differences in LCOH between the Yearly and Hourly reliability scenarios, and between the Hourly and Hourly* scenarios presented in Fig. 3 are noteworthy. Without an accessible grid connection, only tropical or equatorial countries with less seasonality in their solar resource may be able to supply hourly reliable hydrogen without cumbersome overbuild. In higher-latitude

locations, limited and seasonal solar resources would increase the relative attractiveness of fossil-based hydrogen production methods. A grid-connected electricity-based system, however, enables increased flexibility in operations that mitigates the need for a system overbuild even in higher-latitude locations, while maintaining the necessary hydrogen supply reliability that is required at refineries, ammonia plants, liquefaction plants, and other facilities. In fact, we find that the net-zero, hourly-reliable electricity-based hydrogen production pathways that include a grid connection can achieve cost-parity with, if not cost savings compared to, fossil-based alternatives by the next decade (2030 s) in the United States. This assumes that the grid electricity purchased continues to decline in carbon intensity such that the $CO_2$ removal cost for mitigating electric grid-related emissions is marginal. In a future study, it would be worth examining locations outside of the United States with higher grid carbon intensities and relaxed emission reduction goals to see how hourly reliable, net-zero electricity-based pathways with a grid connection fare against fossil-based and off-grid renewable electrolysis alternatives.

In this study we also find that fossil-based production pathways are especially expensive when compared with grid-tied electricity-based solutions in the United States when considering both a cost for removing residual GHG emissions and high upstream natural gas leakage. With these assumptions in place, fossil-based production costs increase over two-fold (see Fig. 4). Therefore, prioritizing best practices with natural gas handling and continued research and development into both methane sensing equipment and negative emission technologies is necessary to maximize the decarbonization potential of, and minimize the cost of, fossil-based hydrogen production solutions into the future. With that said, hourly-reliable electricity-based hydrogen production costs can also be minimized further while maximizing emissions reductions through negative emission technology development and continued advancements in renewable energy and electrolyzer technologies.

From a business point of view, however, net-zero emission hydrogen is less likely to be the lowest cost method for production in the United States and elsewhere given high negative emission technology costs. For now, and most likely for next decade, businesses in the U.S. will take advantage of the 45Q and 45V tax credits described in the Inflation Reduction Act when making hydrogen infrastructure investment decisions. With the 45Q and 45V tax credits included, fossil-based production pathways can remain cost-competitive with electricity-based pathways explored because of the high value of the 45Q tax credit, and the fact that the 45Q tax credit value is independent of the production facility emission intensity or natural gas leakage rate. Fossil-based pathways are especially favorable if embodied emissions of electricity inputs are counted in all production pathways. This makes the decision of which GHG emission accounting scheme to choose for 45V quite consequential for hydrogen developers trying to decide what type of production infrastructure to build out. It is worth noting, however, that jurisdictions like the European Union operate under an emissions trading system (ETS) instead of an incentive-based structure like the United States[31], which makes it difficult for fossil-based hydrogen production to remain viable in these places. Analyzing various hydrogen production pathway costs using policy schemes outside of the United States, such as ETS in the European Union, would be a valuable extension to this work to show what pathways are preferable in different regions of the world.

By the next decade, a grid-connected electricity-based hydrogen production pathway in the United States that takes advantage of 45V approaches, and may surpass, cost-equivalence with fossil-based pathways, with an added marketing advantage of being a lower GHG emitting solution. This is especially true if 45V guidance does not count embodied emissions of renewable energy resources when computing hydrogen production emissions. Incentives aside, fossil-based

hydrogen production pathways can only remain viable in the long-term through life-cycle emission-parity with electricity-based alternatives, thus necessitating the virtual elimination of natural gas supply chain emissions and the use of negative emission technologies to mitigate any remaining emissions. Acting on these challenges, along with continued electricity-based hydrogen production technology development, are integral to scaling a clean hydrogen economy and maximizing emissions reductions in hard-to-abate sectors.

## Methods

The key models used in this study are an electricity-based hydrogen production optimization and a fossil-based hydrogen production computation. The results of these two models are compared to determine which hydrogen production pathways are most optimal from a cost and emission perspective.

### Electricity-based hydrogen system components and optimization structure

Figure 1 details the electricity-based hydrogen production and delivery pathways explored. In our model, we assume hydrogen can either be produced using electricity from an on-site solar PV facility, or from both a solar PV facility and the grid. The grid and PV-generated electricity can either be curtailed, stored in a battery, or used directly to make hydrogen using an electrolyzer. Once hydrogen is produced, it can either be directly delivered to a customer or compressed into a 200 bar storage tank before final end-use. As a sensitivity, we run the model for three locations in the United States with different electricity market conditions and renewable resource potential. These locations are Sacramento, CA, Albany, NY, and El Paso, TX. In the current form of the model, we do not consider space constraints for large system components, such as solar PV or hydrogen storage, but acknowledge this could also impact optimal facility size, production method, and location for a new-build hydrogen production plant.

GHG emissions associated with solar PV and grid electricity are accounted for on a life-cycle basis and are included in the electricity-based hydrogen production cost model. We assume a carbon intensity of 0.04 kg $CO_2$e per $kWh_e$ for embodied solar PV emissions[32,33] and use hourly carbon intensity data derived from NREL's 2035 Cambium Mid-case datasets to represent next decade emissions performance[15]. We modify these hourly emission datasets for our base case scenarios to include embodied emissions from electricity generating sources. Equations and data used to make this adjustment are available in Sections 1 and 3 of the SI, respectively.

We use the Gurobi Linear Optimizer[34] in the Julia[35] mathematical programming tool (JuMP)[36] to size system components such that we minimize the LCOH from electricity-based pathways. Note that linear optimization refers to the structure of the equations within the optimization itself, and not the temporal profile of input data, such as solar electricity production, which is nonlinear over time.

The optimization is subject to the following constraints:
Fixed hydrogen production rate (with varying degrees of reliability)
Fixed solar PV production profile
Operating level < = capacity of each system component
Storage level < = storage capacity
Storage input or output < = storage capacity
Conservation of energy and matter
Conservation of storage levels
Ramping constraints

The full mathematical formulation of this linear optimization can be found in Section 1 of the SI[24]. Note that the optimization objective function target is to minimize the LCOH of hydrogen produced from electricity-based pathway in $/kg $H_2$ produced. This is found by dividing the optimal annualized hydrogen production cost by the total amount of hydrogen produced and delivered to a co-located hydrogen consumer.

**Table 2 | Electricity-based hydrogen production model main input parameters**

| Parameter | Next Decade Value | Units | Source |
|---|---|---|---|
| Hydrogen Supply | 250 | metric ton/day | N/A |
| Solar PV Capacity Factor (hourly) | 27.5 (CA) 20.0 (NY) 29.6 (TX) | % average | 16 |
| Capital Cost of Electrolyzer | 460 | $/$kW_e$ (100 MW system) | 19, 39, 40 |
| Capital Cost of Solar Farm | 600 | $/$kW_e$ (100 MW system) | 19, 37, 38 |
| Capital Cost $H_2$ Storage | 500 | $/kg $H_2$ | 19, 48 |
| Capital Cost Battery Storage | 250 | $/$kWh_e$ | 19, 38, 49 |
| Project Life | 25 | years | 19 |
| WACC (Discount Rate) | 8 | % | 19 |
| Grid Electricity Price (hourly) | 0.033 (CA) 0.036 (NY) 0.023 (TX) | $/$kWh_e$ yearly average | 50 |
| Grid Emissions (hourly w/ embodied) | 0.077 (CA) 0.038 (NY) 0.077 (TX) | kg $CO_2$/$kWh_e$ average | 15, 33, 51 |
| Grid Demand Charge | 10 (CA) 5 (NY) 5 (TX) | $/max $kW_e$/month | 28 |
| Solar PV Life-cycle Emissions | 0.04 | kg $CO_2$/$kWh_e$ | 32 |
| Emissions Removal Cost | 200 | $/metric ton $CO_2$e removed | 17–19 |

We assume all costs are in 2020 dollars. See the SI, Section 3 for all input parameter values. PV refers to photovoltaic. WACC refers to weighted average cost of capital.

The primary data values used in the optimization are shown in Table 2 with the complete data available in Section 3 of the SI[24]. Notably, a cost of $200 per metric ton $CO_2$e is chosen for removing any $CO_2$e emissions from electricity input[17–19]. This cost is meant to be representative of a next decade negative emission technology (e.g., direct air capture with carbon capture and storage (CCS), bioenergy with CCS, enhanced weathering, soil carbon enhancement). Because it is not our goal to evaluate these removal costs in detail in this paper, we choose a nominal "technology agnostic" value ($200/metric ton) and apply it evenly across all $H_2$ pathways. See sensitivity analysis for the impact of choosing other $CO_2$ removal costs.

We assume next-decade capital costs for the solar PV plant and the electrolyzer are $600/kW (100 MW system)[37,38] and $460/kW (100 MW system)[39,40], respectively, with these including both balance of plant and installation costs. Due to technology learning, the solar PV and electrolyzer next decade capital costs are 66% and 52% of their current capital costs, respectively (see Section 3 of SI)[37–40]. We include a capital cost scaling factor for both electrolyzer and solar PV build-out in our model to account for economies of scale, separately from next decade technology learning costs. Typically, cost scaling factors are between 0.6 and 0.8 but we choose scaling factors of 0.95 and 0.9 for electrolyzers and solar PV facilities, respectively, as our central case. This is because it is difficult to disentangle economies of scale from technology advancement when analyzing cost projections, and the cost estimates we include already account for next-decade technology learning curves. Economies of scale for a given plant are considered using Equation #1.

$$C_s = C_B \left( \frac{S_s}{S_B} \right)^m \qquad (1)$$

where:

$C_s$ = scaled cost

$C_B$ = baseline cost

$S_s$ = scaled hydrogen production rate

$S_B$ = baseline production rate

$m$ = capital cost scaling factor

When modeling electricity-based pathways that include a grid interconnection, our next-decade cases use state-wide hourly marginal electricity pricing data from NREL's Cambium database[15]. Our baseline results in Figs. 3, 4 use the 2035 Cambium Mid-case scenario electricity price data. For sensitivity analyses shown in Figs. 4, 5, we also include the Low Renewable Energy Cost and High Renewable Energy Cost scenarios from Cambium. Excess electricity generated from the solar PV plant, or pulled from the grid, can be stored in a battery, and sold back to the grid during hours when these marginal prices are high if it means savings to the LCOH overall. The model also accounts for demand charges and a grid interconnection fee that the hydrogen producer will have to pay[19,28,41]. It is important to note that there is a high level of uncertainty in future marginal prices and demand charges across the United States and the next-decade electricity tariffs chosen in this study reflect only a few of many possible outcomes. We also do not account for how incremental electricity demand for hydrogen production facilities will themselves impact marginal electricity prices by the next decade. Developing a stochastic optimization, based on probabilistic electricity prices and imperfect information, could be a valuable extension to this work, but is beyond the scope of this paper. Lastly, it should be mentioned that average marginal electricity prices shown in Table 2 do not reflect the average price of grid electricity used in grid-tied pathways since the optimization model chooses to use power in subsets of hours depending on solar availability, electricity pricing, and the reliability target in place.

## Fossil-based hydrogen system components and data

Figure 2 shows the fossil-based hydrogen production and delivery pathways explored in this study. In our fossil-based hydrogen modeling, we assume hydrogen is produced from either a steam methane reformer (SMR) or an auto-thermal reformer (ATR). The energy inputs to the reformer include grid electricity and natural gas. Once hydrogen is produced, it is directly delivered to the consumer. We assume the reformer operates at 90% capacity factor, with a near-constant production rate to achieve hourly delivery reliability necessary for off takers such as refineries and ammonia production plants.

The carbon dioxide generated at the reformer is directly emitted to the atmosphere in some production pathways, but is also partially captured, transported, and geologically sequestered in other pathways. Carbon dioxide capture pathways explored include flue gas carbon capture for an SMR-CCS (1), flue gas capture and process-based capture for an SMR-CCS (2), and process-based capture for an ATR-CCS (3). Each of these capture schemes results in a different carbon capture efficiency. For the purposes of this study, we assume minimal $CO_2$ transport will be required and $CO_2$ storage will cost $10 per metric ton $CO_2$ stored[42].

Other GHG emission considerations for the fossil-based hydrogen production pathways include grid electricity and natural gas life-cycle emissions. We use the same grid electricity emissions data for the fossil-based hydrogen production model as the electricity-based hydrogen production model. For upstream natural gas emissions, excluding natural gas leakage, we assume a carbon intensity (CI) of 0.3 kg $CO_2$e per kg $CH_4$[43]. Finally, given the uncertainty and locational variation of natural gas leakage[44], we perform emission sensitivity analysis for several natural gas leakage rates (up to 4%) as well as global warming potential timeframes (100-year or 20-year GWP timeframes). The global warming potential, or GHG potency relative to $CO_2$, for natural gas is higher for a 20-year timeframe than a 100-year timeframe because of the short-lived nature of natural gas in the atmosphere[45].

The primary input values used for each of the four fossil-based production pathways are shown in Table 3 with complete data available in Section 4 of the SI[24]. The majority of input data is derived from the National Energy Technology Laboratory's (NETL's) recent report comparing the cost and emissions of various fossil-based hydrogen production pathways[22]. Natural gas and electricity price inputs, however, are location-specific[15,46]. A capital cost scaling factor of 0.6 is used to consider economies of scale for fossil-based production pathways. This is the same scaling factor used in NREL's hydrogen production cost models (H2A)[47]. Note that this is a lower capital cost scaling factor than used for the electricity-based pathways because fossil-based pathways are more mature with discernable economies of scale.

**Table 3 | Fossil-based hydrogen production main model input parameters**

| Input Parameters | Next Decade Value | Units | Source |
|---|---|---|---|
| $CO_2$ Capture Percent | 0% [SMR]<br>56% [SMR-CCS (1)]<br>96% [SMR-CCS (2)]<br>95% [ATR-CCS (3)] | % | 22 |
| Electricity Use | 0.65 [SMR]<br>1.5 [SMR-CCS (1)]<br>2.04 [SMR-CCS (2)]<br>4 [ATR-CCS (3)] | kWh$_e$/kg $H_2$ capacity | 22 |
| Natural Gas Use | 3.53 [SMR]<br>3.58 [SMR-CCS (1)]<br>3.75 [SMR-CCS (2)]<br>3.52 [ATR-CCS (3)] | kg $CH_4$/kg $H_2$ capacity | 22 |
| Capital Costs | 549 [SMR]<br>734 [SMR-CCS (1)]<br>1336 [SMR-CCS (2)]<br>1056 [ATR-CCS (3)] | $/kW $H_2$ capacity | 22 |
| WACC (Discount Rate) | 5% | % | 22 |
| Natural Gas Cost | $6.50 [CA]<br>$5.00 [NY]<br>$4.00 [TX] | $/MMBTU $CH_4$ | 46 |
| Emissions Removal Cost | 200 | $/metric ton $CO_2$e removed | 19 |

(1) Refers to an SMR with process $CO_2$ capture. (2) Refers to an SMR with process and flue gas $CO_2$ capture. (3) Refers to an ATR with process $CO_2$ capture. We assume all costs are in 2020 dollars. See the SI, Section 4 for all input parameter values. SMR refers to steam methane reforming. CCS refers to carbon capture and storage. ATR refers to auto-thermal reforming. WACC refers to weighted average cost of capital.

## Data availability
All input and results data used to generate main text figures are available in a GitHub repository[25]. Source data are provided with this paper in the repository. All other input data is found in the supporting information, which can also be obtained on GitHub[25]. Source data are provided with this paper.

## Code availability
The hydrogen production cost models and supporting code are available in the same Github repository[25]. The mathematical formulation of the electricity-based and fossil-based models are also found in the GitHub repository as well as in this study's supporting information.

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

## Acknowledgements

This work was funded by the Stanford Natural Gas Initiative, an industry consortium that supports independent research at Stanford University. The authors would like to thank Jacques de Chalendar and Jabs Aljubran for their assistance with data collection.

## Author contributions

J.M.B., E.D.S., A.R.B., and N.L.B. conceptualized the study. J.M.B. developed the cost model and applied the model to this study. J.M.B. was advised by both E.D.S. and A.R.B. during model development and implementation stages. E.D.S. and A.R.B. contributed datasets. J.M.B. drafted and finalized the manuscript. E.D.S., A.R.B., and N.L.B. advised on analysis and revised the manuscript.

## Competing interests

The authors declare no competing interests.
