## [Peer Review File · Nature Communications]

A Cost Comparison of Various Hourly-Reliable and Net-Zero Hydrogen Production Pathways in the United StatesREVIEWER COMMENTS

Reviewer #1 (Remarks to the Author):

This manuscript is a good piece of work, dealing with known technologies and related unsolved question marks regarding operation and effectiveness. The study tries to offer a good "fair comparison" point of view. Structure is good, development is clear, conclusions are supported and sound.

I have a few requests and suggestions to improve the content and the dissemination. The high number of points below is not a reflection of a high number of issues, but rather of an effort to aid authors towards improving the work. The main needs are: add quantitative elements in the text, consider larger sensitivity ranges on some parameters, and consider the addition of one grid-connected but also constrained case.

- In the Introduction, line 47, authors are a little too strong in saying that "no work is looking at this". You may have a look at literature (e.g., a few in Int J Hydrogen Energy or Energy Conversion & Management), where some works exist on hourly-resolved analysis of Power-to-Hydrogen systems.
- In line 57, authors refer to "energy storage" which is probably meant as "electrical energy storage".
- In line 66, the base cost of 200 \$ per metric tonne of CO₂ is proposed. Please note that this is quite low for today and next-future when thinking of DAC. Negative emissions via BECCS are correct to be mentioned, but also very highly requested as compensation by many processes and not available in very large scale.
- In line 76, when introducing the incentive schemes active in the USA, authors should spend a few lines to provide a brief overview on them, in order to aid non-USA readers' understanding. This might equally be located elsewhere in the main text.
- The considered plant scale could be anticipated in the Introduction.
- In Figure 1a there are two H₂ output lines. According to the text, there is only one delivery point, which extract from either the electrolyser or the storage. Please reroute the streams into one unique plant outlet.
- There is no representation in Figure 1a nor mention in the text of water purification needs (in terms of both equipment and energy consumption). Also, being California highly susceptible to droughts, water supply should be discussed and possibly water needs computed for the different pathways.
- At this point in the manuscript, it is not clear what is included in "upstream solar PV CO₂ emissions" and the use of LCA vs WTW approach.
- In line 103, the plant scale is declared. (1) Why is such number selected? (2) It would be useful for the reader to be given a reference (typical H₂ production output of a refinery, nominal hydrogen production of a 10 MW electrolyser, ...).
- Line 108, again "upstream solar PV CO₂ emissions" are declared but not specified. Sometimes along the text it seems that LCA and WTW approaches are mixed up.
- Please add a few lines and/or a table with main techno-economic input data to aid the understanding without the need to access SI. In particular, the relative specific investment cost of electric energy storage and of hydrogen tanks is essential to drive the sizing of the two in the optimization. Also, please state that complete data are available in SI.
- Line 120 and elsewhere. The case of grid-connected PV+electrolysis operation is extremely relevant and often underrated, the presence of this option in the analysis is highly appreciated. For a complete understanding, it is required to provide the share of grid electricity input over the total electricity input (PV+grid) to the electrolyzer; also, the share of PV electricity that is not used by the electrolyzer (either curtailed or sent to electric grid) should be provided in both "pv" and "pv+grid" cases.
- Also, it would be useful to offer the information of battery and H₂ tank sizing in terms of days equivalent of nominal production.
- Line 124, what is the "high" penetration expected? Please be more quantitative.
- Line 129, "if grid prices shift to...". Please be more quantitative. The case of profoundly different electricity market prices is very likely in the future as soon as solar (and/or wind) will be covering a very large share of generation in some hours of the day.

- Lines 146 and following. The point of natural gas leakage importance was strongly stated also in XX. Might be a useful ref to add.
- Line 150. Again, please discuss the share of grid electricity on total electricity used by electrolyser. In the EU, the directives are getting stricter and stricter on co-location and contemporaneity of RES power plants and electrolysis units (see https://ec.europa.eu/commission/presscorner/detail/en/qanda_23_595 and updates).
- What happens if a lower share (e.g., max 10% or 20% of grid input over the year) is imposed? This would be a relevant case to add.
- Line 155: include the CO2 capture rates assumed in the three cases.
- Line 196 and followings. This is a very nice finding!
- The electricity purchase price appear in figure 5 and not in the text. The upper limit of the sensitivity is 84 \$/MWh, which is not very high: Europe has seen peak prices above 700 €/MWh throughout 2022. Hopefully these will not repeat and will not be the annual average, but given that the work is doing a sensitivity assessment, the range could be enlarged. The same goes for natural gas price, which has now long remained above 50 €/MWh_HHV (peaked over 200 €/MWh_HHV).
- Line 222, the reason of varying between including and excluding PV upstream emissions should be described. Also, what is included in those upstream LCA should be clarified. Why is it not done for grid electricity, whose emission factors are assumed very low? (remember that a good NG-fired CCGT has CO2 emissions in the order of 350-400 gCO2/kWh just for the combustion at facility point).
- Line 226. Good point, agreed.
- Line 279 and following. This is true in the USA system with current schemes. But not in a system where a "carbon tax" is applied on CO2 emissions, such as the ETS scheme active in the EU. Might be interesting to deepen/mention.
- Line 305. Hydrogen compression is mention, which never appeared before. Why is that? Where is it located? It is not depicted in Figure 1. Is it accounted for in capital and operational costs? What is the considered pressure? ...
- Line 309: 50% reduction of CO2 emissions in a future grid where RES power plants are massive and energy storage are necessarily significant might not be in line with the LCA approach, which would require to include upstream emissions for those installations... Please verify the assumption and/or discuss it.
- Line 365. A revenue of 56 \$/MWh is quite high (larger than expected LCOE) and may not be true in a future high-RES electric system where the overgeneration of our plant will likely match with the overgeneration of many other plants. The analysis should include a case that consider zero remuneration of surplus generation (i.e., it is curtailed and not sold to grid).
- Lines 376 and following. Please summarize reforming assumptions in a table (see also comment for techno-economic input data above).

Reviewer #2 (Remarks to the Author):

Leveling the Playing Field: A Cost Comparison of Various Hourly-Reliable, Net-Zero Hydrogen Production Pathways
Submitted to Nature Communications

In this manuscript, the authors investigate the future hydrogen costs with different production scenarios, namely electricity-based hydrogen with and without a grid, fossil-based hydrogen with different carbon capture methods. By comparing the cost components, the study aims to outline the development pathways for achieving net-zero hydrogen production. In addition, the impact of government incentive schemes has been investigated. Optimization models and simulation models have been used, with the support of data focusing on Sacramento and California region. The study highlights the importance of grid electricity in electricity-based hydrogen production. In addition, with US Inflation Reduction Act, the fossil-based production could be in favor, which could be against the initial motivation of the incentive scheme.

I have some major concerns, which need authors' consideration.

Major criticisms

1. There are quite many results of the study, often valid with the specific conditions of the study. But it is still unclear what are commonly known, what are the new findings in current study. A better presentation is needed to see the value of the study.

2. One interesting aspect of the study is the pathway of hydrogen development, which also means that the evolving path of the cost components could be interesting. As the authors have the "Current, Next decade and Mid-century values" of input parameters, would it be possible to describe the development of the hydrogen costs?

3. I agree with the authors that grid electricity is important for hydrogen production, and also I believe that hydrogen production should be connected to grids, at least in the initial stage. But there are some issues which could be ignored by the authors.

- The electricity price in the grid should be stochastic, as long as it is traded in an open market. This is the case in many countries, and I assume the same in California. However, the model (Section 1 of the SI) seems only considering one price scenario (perhaps hourly price in one year). If this is true, would it be sufficient?

- The electricity price may have a strong change in the future, especially if the electricity production sources are developed into renewables such as solar and wind. The intermitted supply eventually could make the grid price with a very high volatility (when marginal cost is used for pricing), which will then have a significant impact on the grid-based hydrogen. As the study focuses on the hydrogen pathway, such impacts in the long-run should be relevant.

- I am not sure what is the scale of electricity-based hydrogen production compared with the local electricity market. If it is sufficiently large, hydrogen production could affect electricity price, which will again affect the hydrogen price. This may not be an issue currently but could be a problem when hydrogen market is large.

On lines 366-367, there are some related discussions. But the above concerns with electricity need to be more clearly elaborated.

4. The conclusion of the study is rather based on the specific location Sacramento and California. We then need to be more careful in interpreting the results. For example, the solar capacity factor is 31% (see Section 3 of the SI), this is much higher than the average value in other regions worldwide, which is typically 10%-20%. It might also be good to present what the major sources of electricity production in California are (current and future), in order to understand the grid carbon intensity values (lines 310-311).

5. I am not quite sure why a linear optimization approach can be used for defining the capacity of solar PV, battery, electrolyzer and hydrogen storage. If we see electricity supply from solar PV, it should be random and follow a pattern of sine function. This means that the installed solar PV capacity and storage capacity should follow a non-linear relationship. Seasonal variation adds more complexity, and similar situation is applied to the hydrogen storage when the electrolyzer is restricted to the solar PV inputs. The authors also have similar concerns on lines 98-101, but the model is not reflecting this non-linear condition.

6. Conclusion: "A key finding from this study is that net-zero electricity-based hydrogen production with variable energy input becomes increasingly expensive with stricter reliability constraints and no grid connection." This seems quite trivial to me for a solar PV and electrolyzer system, as the CF of solar PV has restricted the input to electrolyzer and therefore the capital costs are high in such a setting.

Other comments

7. Lines 105-109 (also figure 3): "...yearly reliable hydrogen delivery is lowest cost, with total LCOH more than doubling as reliability constraints increase to the hourly timeframe". This is not surprising, as matching demand and supply is more difficult on the hourly basis, which will lead to overbuilt (over capacity of the electrolyzer and solar PV).

8. Lines 120 and 122: "...a grid connection allows for hourly delivery reliability at a 45% lower cost ..."

This is also not surprising, as the grid electricity now is a consistent feedstock to the electrolyzer, coupling the capacities of solar PV and electrolyzer is less important, and overbuilt is not necessary.

9. Lines 307-308: "a carbon intensity of 0.04 kg CO₂e per kWh for upstream solar PV emissions ..." I am not sure of this. Since the upstream emission is mainly due to the equipment of Solar PV, the measure should base on per kW installed capacity, not on energy produced?

10. Lines 340-341: "... the solar PV plant and the electrolyzer are \$500/kW (100MW system) and \$570/kW (1 MW system) ..." I understand the investment cost will include the scale factor. But still the above 100MW and 1MW systems seem not comparable. Some explanation?

11. Section 1 in the SI: I interpret E_U as the electricity produced by solar PV and retained in the system (electrolyzer, compressor, battery), according to line 169. If this is correct, the constrain on line 177 should be revised: the grid electricity should not be reduced on the left of the equation, as it together with solar PV output will be the input for the electrolyzer.

12. Section 1 in the SI: What is the decision principle of curtailed electricity? I think when solar PV generates more electricity the input to be consumed by the electrolyzer system (including battery etc.), the rest is sent to the grid according to the balance constraints. But if the electricity grid price is high, can we send more electricity to the grid? Not quite clear about the decisions here.

13. Section 3 in the SI: I interpret Hydrogen Energy Content (33.3 kWh/kg) as electricity consumption per kg hydrogen production. Please check this value, I have the value which is about 49kWh/kg H₂.

To summarize, the study presents a very interesting topic and urgent issues. The study results could be very important for guiding the decision in energy transition, in particular for the regions considering hydrogen as a future alternative. However, data and analysis part need some further explanation and clarification to ensure the results. In additional, I believe that the presentation of results and conclusions can be improved, to indicate what are the main findings, as well as in which circumstances the findings are valid. I would therefore suggest a major revision of the manuscript.

Reviewer #3 (Remarks to the Author):

The manuscript addresses a topical problem in the energy sector around the hydrogen economy. I have a few comments.

As also highlighted in the manuscript, there have been numerous models introduced for investment decision analysis in various hydrogen vectors. The paper places its emphasis on the time resolution of past studies and argues that the granularity of the planning horizon (e.g. hourly profile) increases the decision accuracy. Indeed a valid point, but not an unknown fact. Consideration of any energy storage in planning requires a granular timeframe. However, if this is a limitation, there are other limitations as well that the manuscript doesn't address. For instance, seemingly the authors are using one capacity factor for solar output which significantly varies over locations. The use of solar+storage vs solar+storage+grid though is useful, doesn't add anything new to the current understanding.

I have also concern about the results shown in Figure 3. The LCOH for reliability at hourly, daily, and monthly levels have relatively close values (say 5.5-6.5), but on a yearly basis, it suddenly drops to 2.5. This sharp decline from a monthly base to a yearly base is suspicious.

Last but not least, despite its generic title, this manuscript is limited to specific US policy scenarios.

Reviewer #1:

Author responses below are in **bolded** text, with updates to the manuscript reproduced in **blue bolded** text.

This manuscript is a good piece of work, dealing with known technologies and related unsolved question marks regarding operation and effectiveness. The study tries to offer a good "fair comparison" point of view. Structure is good, development is clear, conclusions are supported and sound.

I have a few requests and suggestions to improve the content and the dissemination. The high number of points below is not a reflection of a high number of issues, but rather of an effort to aid authors towards improving the work. The main needs are: add quantitative elements in the text, consider larger sensitivity ranges on some parameters, and consider the addition of one grid-connected but also constrained case.

We agree with this assessment and have updated the manuscript to address each of these three key points raised, which we described in detail in our responses below.

- In the Introduction, line 47, authors are a little too strong in saying that "no work is looking at this". You may have a look at literature (e.g., a few in Int J Hydrogen Energy or Energy Conversion & Management), where some works exist on hourly-resolved analysis of Power-to-Hydrogen systems.

This is true. The statement has been revised to mention some of these studies while also noting that these studies that do include hourly-derived LCOH values do not include an emission-adjusted cost.

Manuscript (Lines 47 – 50):

“In many prior studies, reliability distinctions are either unclear or not addressed [1]–[5], and studies that do focus on hourly-resolved power-to-hydrogen systems tend to leave out emissions-adjusted cost comparisons with conventional fossil-based hydrogen production [6]–[8].”

- In line 57, authors refer to "energy storage" which is probably meant as "electrical energy storage".

The optimization can choose between battery storage or hydrogen storage. We have updated the manuscript to eliminate the ambiguity.

Manuscript (Lines 62 – 65):

“All electricity-based production pathways explored in this study consider an onsite-solar photovoltaic (PV) facility with the option to include energy storage (battery or compressed hydrogen storage at 200 bar) and/or a grid interconnection to increase delivery reliability.”

- In line 66, the base cost of 200 \$ per metric tonne of CO₂ is proposed. Please note that this is quite low for today and next-future when thinking of DAC. Negative emissions via BECCS are correct to be

mentioned, but also very highly requested as compensation by many processes and not available in very large scale.

200 \$/MT is chosen based on Climeworks' target DAC costs in the medium-term. Sentence updated to include source.

Manuscript (Lines 83 – 85):

“All scenarios assume a next-decade emission mitigation cost of \$200/metric ton CO₂e [9]–[11], representing a technology agnostic mitigation cost as the details of offsite mitigation are beyond the scope of this paper...”

- In line 76, when introducing the incentive schemes active in the USA, authors should spend a few lines to provide a brief overview on them, in order to aid non-USA readers' understanding. This might equally be located elsewhere in the main text.

More detail on these incentive schemes are provided later in the manuscript when discussing the impact IRA has on the LCOH of each hydrogen production pathway explored.

Manuscript (Lines 305 – 312):

“With the passing of the Inflation Reduction Act in the United States, we separately consider the effect of 45V and 45Q tax credits. These crediting schemes operate differently compared to the above approach of applying an emission mitigation cost to all emissions such that the resulting hydrogen is net-zero. As stated in section 45V of the Inflation Reduction Act [12], hydrogen production pathways that have life-cycle emissions less than 0.45 kg CO₂e/kg H₂ can receive \$3.00/kg H₂ PTC for 10 years, while pathways with emissions between 0.45 and 1.5 kg CO₂e/kg H₂ can receive a \$1.00/kg H₂ PTC, pathways between 1.5 and 2.5 kg CO₂e/kg H₂ receive a \$0.75/kg H₂ PTC, and pathways between 2.5 and 4 kg CO₂e/kg H₂ receive a \$0.60/kg H₂ PTC for 10 years of operation. Fossil-based hydrogen producers have the flexibility to choose between 45V or 45Q, with 45Q providing the hydrogen producer a tax credit of \$85/metric ton CO₂ permanently captured and sequestered for 12 years of operation.”

- The considered plant scale could be anticipated in the Introduction.

Production plant sizing is now mentioned in the Introduction.

Manuscript (Lines 57 – 60):

“We set the production facility size to 250 metric ton per day (roughly 500 MW for electrolysis), a typical size of hydrogen production plants at refineries [14], to reflect a next-decade future with growing hydrogen demand and economies of scale benefits.”

- In Figure 1a there are two H₂ output lines. According to the text, there is only one delivery point, which extract from either the electrolyser or the storage. Please reroute the streams into one unique plant outlet.

This is a very good point. Thank you for bringing this up. Figure 1 has been updated.

Manuscript (Lines 105 – 106)

- There is no representation in Figure 1a nor mention in the text of water purification needs (in terms of both equipment and energy consumption). Also, being California highly susceptible to droughts, water supply should be discussed and possibly water needs computed for the different pathways.

We now mention geographic variability in water access and water quality in the Introduction. We choose not to focus on this topic too much, however, given that water is only a few cents addition to the LCOH.

Manuscript (Lines 62 – 68):

“All electricity-based production pathways explored in this study consider an onsite-solar photovoltaic (PV) facility with the option to include energy storage (battery or compressed hydrogen storage at 200 bar) and/or a grid interconnection to increase delivery reliability. Water use is also considered, although, we do not as far as to consider how this cost changes by location given geographic variability in water access and water quality.”

- At this point in the manuscript, it is not clear what is included in "upstream solar PV CO2 emissions" and the use of LCA vs WTW approach.

Updated manuscript in Introduction section to explain emissions accounting in more detail.

Manuscript (Lines 68 - 71):

“GHG emissions from electricity input are accounted for on a life-cycle basis; meaning embodied emissions (emissions from materials extraction, manufacturing, construction, etc.) for solar PV and other grid electricity sources are considered in the electricity-based pathways. See Figure 1 for further detail on the system configuration as well as emissions considerations...”

Manuscript (Lines 75 - 79):

“GHG emissions accounted for in the fossil-based pathways include those directly emitted from the production plant as well as life-cycle emissions (including embodied emissions) of the input electricity and natural gas. See Figure 2 for more detail on the fossil-based hydrogen production system configuration and GHG emission inclusions.”

- In line 103, the plant scale is declared. (1) Why is such number selected? (2) It would be useful for the reader to be given a reference (typical H2 production output of a refinery, nominal hydrogen production of a 10 MW electrolyser, ...).

Good points! This has been addressed in the updated manuscript in the introduction.

Manuscript (Lines 57 - 60):

“We set the production facility size to 250 metric ton per day (roughly 500 MW for electrolysis), a typical size of hydrogen production plants at refineries [13], to reflect a next-decade future with growing hydrogen demand and economies of scale benefits.”

- Line 108, again "upstream solar PV CO2 emissions" are declared but not specified. Sometimes along the text it seems that LCA and WTW approaches are mixed up.

Manuscript has been updated to specify that any energy input to a facility has emissions measured using life-cycle or embodied approach (e.g. solar PV emissions include mining materials, PV construction, end-of-life, etc.)

Manuscript (Lines 68 - 71):

"GHG emissions from electricity input are accounted for on a life-cycle basis; meaning embodied emissions (emissions from materials extraction, manufacturing, construction, etc.) for solar PV and other grid electricity sources are considered in the electricity-based pathways. See Figure 1 for further detail on the system configuration as well as emissions considerations..."

- Please add a few lines and/or a table with main techno-economic input data to aid the understanding without the need to access SI. In particular, the relative specific investment cost of electric energy storage and of hydrogen tanks is essential to drive the sizing of the two in the optimization. Also, please state that complete data are available in SI.

Updated manuscript has a Table 1 in the methods section that contains the primary input data for the optimization. At the end of the Introduction, we also note the following:

Manuscript (Lines 101 - 104):

"See Methods section and supporting information (SI) for a detailed discussion of modeling methods and input assumptions. Tables 1 and 2 in the methods section list each of the primary input values used in the electricity- and fossil-based cost models, respectively. All other input values are provided in the SI."

- Line 120 and elsewhere. The case of grid-connected PV+electrolysis operation is extremely relevant and often underrated, the presence of this option in the analysis is highly appreciated. For a complete understanding, it is required to provide the share of grid electricity input over the total electricity input (PV+grid) to the electrolyzer; also, the share of PV electricity that is not used by the electrolyzer (either curtailed or sent to electric grid) should be provided in both "pv" and "pv+grid" cases.

Completed in updated manuscript by providing results for the California location. Additional operational information is presented in the Supplemental Information Section 9.

Manuscript (Lines 145 - 148):

"For example, in the Yearly case for California, only 2.1% of electricity produced from the solar PV facility goes unused by the electrolyzer, but as reliability constraints shift to monthly, daily, or hourly levels, the percent of solar PV electricity that is curtailed increases to 31%, 38%, and 39%, respectively."

Manuscript (Lines 156 - 162):

“The *Hourly** case has unlimited access to the grid while the *Hourly*** case is constrained to 10% electricity usage from the grid for hydrogen production operations. As shown, an unconstrained grid connection allows for hourly delivery reliability at a 48-69% lower cost compared to the *Hourly* pathway, depending on the state. We also notice in the *Hourly** case that electricity input for production comes almost exclusively from the grid due to near-zero marginal pricing during hours when excess renewables would otherwise be curtailed [14].”

- Also, it would be useful to offer the information of battery and H2 tank sizing in terms of days equivalent of nominal production.

Storage sizes included in updated manuscript in terms of metric ton H2 equivalence.

Manuscript (Lines 148 - 151)

“In terms of storage requirements, the *Yearly* case requires no storage, while the *Monthly*, *Daily*, and *Hourly* pathways require 2,030 metric ton, 2,330 metric ton, and 2,500 metric ton of H2 storage, respectively. Battery storage installment is minimal for each reliability case due to the lower relative cost of storage using hydrogen tanks (see Table 1 for cost data).”

Manuscript (Lines 167- 173)

“In this location, we find only 92 metric tons and 210 metric tons of hydrogen storage are used in the *Hourly** and *Hourly*** pathways, respectively. These are an order of magnitude lower than the storage in the off-grid *Hourly* case (2,500 metric tons). However, these two grid-tied pathways also take advantage of small battery storage systems, equivalent to 11 metric tons hydrogen storage (580 MWh) in the *Hourly** pathway and 13 metric ton (680 MWh) in the *Hourly*** pathway, to store excess solar PV electricity and low-price grid electricity for later use.”

- Line 124, what is the "high" penetration expected? Please be more quantitative.

This sentence of the manuscript has been taken out.

- Line 129, "if grid prices shift to...". Please be more quantitative. The case of profoundly different electricity market prices is very likely in the future as soon as solar (and/or wind) will be covering a very large share of generation in some hours of the day.

The analysis in this section has been redone because we found new hourly electricity price forecast data from NREL's Cambium database [14]. We use the 2035 Mid-Case electricity price data from Cambium for our base case results shown in the manuscript. This replaces the electricity price data we were originally using which was based on 2020 marginal prices. We made this change to make sure our LCOH results best reflect next-decade conditions. We also look at low renewable energy cost and high renewable energy cost scenarios from Cambium as sensitivities. The electricity price sensitivities are shown in Figure 5. See below for updated sections that discuss the electricity price data used.

Manuscript (Lines 65 - 66)

“Grid electricity prices are derived from NREL's Mid-Case scenario in the Cambium database [14].”

Manuscript (Lines 251 - 259)

“To capture uncertainty and variability in grid electricity prices, we use the National Renewable Energy Laboratory’s (NREL’s) 2035 Cambium Mid-Case as our central scenario and the Low Renewable Energy Cost and High Renewable Energy Cost scenarios as the low and high price bounds, respectively [15]. We find using these electricity price scenarios does not result in as large of a change in LCOH as that of the grid demand charge. However, on a relative basis, the LCOH is more sensitive to average electricity price than it is to grid demand charge. This finding is shown in Figures S.1 of the SI which contains spider plots for the PV/Storage/Grid* production pathway. Figure S.2 contains similar spider plots, but for the ATR-CCS production pathway.”

Manuscript (Lines 173 - 179)

“The average price of electricity sold back to the grid in the *Hourly** and *Hourly*** pathways amounts to \$1,130/MWh and \$66/MWh, respectively, while the average price of electricity bought from the grid in these cases equals \$19/MWh and \$24/MWh, respectively. These price differences show how important the interaction between the grid and battery storage system can be for producing low-cost, electricity-based hydrogen. Again, these trends remain consistent for the Texas and New York production locations. Explore Section 9 of the SI for more detail on electricity-based hydrogen production operations.”

- Lines 146 and following. The point of natural gas leakage importance was strongly stated also in XX. Might be a useful ref to add.

Reference added.

Manuscript (Lines 209 - 211)

“The fossil-based production pathways become especially uneconomic from a net-zero perspective when upstream natural gas leakage exceeds 4%. This point on the significance of natural gas leakage has been highlighted in other studies as well [15], [16]”

- Line 150. Again, please discuss the share of grid electricity on total electricity used by electrolyser. In the EU, the directives are getting stricter and stricter on co-location and contemporaneity of RES power plants and electrolysis units (see https://ec.europa.eu/commission/presscorner/detail/en/qanda_23_595 and updates).

Completed. See discussion from the manuscript below.

Manuscript (Lines 156 - 166)

“The *Hourly** case has unlimited access to the grid while the *Hourly*** case is constrained to 10% electricity usage from the grid for hydrogen production operations. As shown, an unconstrained grid connection allows for hourly delivery reliability at a 48-69% lower cost compared to the *Hourly* pathway, depending on the state. We also notice in the *Hourly** case that electricity input for production comes almost exclusively from the grid due to near-zero marginal pricing during hours when excess renewables would otherwise be curtailed [14]. While not as dramatic, the *Hourly***

pathway also results in cost savings compared to the off-grid *Hourly* case, with an LCOH reduction of 42% in California, 19% in Texas, and 43% in New York. These cost savings are realized because an overbuild of solar PV and energy storage is no longer necessary and excess on-site electricity can be sold back to the grid when prices are high instead of being curtailed.”

- What happens if a lower share (e.g., max 10% or 20% of grid input over the year) is imposed? This would be a relevant case to add.

We have added a case with a constraint of 10% electricity input from the grid. We see in this case that costs are higher than the *Hourly** (unconstrained grid use) case because the facility has less ability to use the near-zero cost electricity from the grid. In the *Hourly** case, almost all electricity comes from the grid, as can be seen in Figure 3. In the *Hourly*** case, grid use is maximized in each situation. See below for sections of the manuscript where we discuss the *Hourly** (PV/Storage/Grid**)* case.

Manuscript (Lines 155 - 179)

“The two rightmost stacked bar charts in each subplot of Figure 3 represent hourly reliability electricity-based production pathways that also have an accessible grid connection. The *Hourly** case has unlimited access to the grid while the *Hourly*** case is constrained to 10% electricity usage from the grid for hydrogen production operations. As shown, an unconstrained grid connection allows for hourly delivery reliability at a 48-69% lower cost compared to the *Hourly* pathway, depending on the state. We also notice in the *Hourly** case that electricity input for production comes almost exclusively from the grid due to near-zero marginal pricing during hours when excess renewables would otherwise be curtailed [15]. While not as dramatic, the *Hourly*** pathway also results in cost savings compared to the off-grid *Hourly* case, with an LCOH reduction of 42% in California, 19% in Texas, and 43% in New York. These cost savings are realized because an overbuild of solar PV and energy storage is no longer necessary and excess on-site electricity can be sold back to the grid when prices are high instead of being curtailed.”

“To support this finding, we again examine operations for the California location. In this location, we find only 92 metric tons and 210 metric tons of hydrogen storage are used in the *Hourly** and *Hourly*** pathways, respectively. These are an order of magnitude lower than the storage in the off-grid *Hourly* case (2,500 metric tons). However, these two grid-tied pathways also take advantage of small battery storage systems, equivalent to 11 metric tons hydrogen storage (580 MWh) in the *Hourly** pathway and 13 metric ton (680 MWh) in the *Hourly*** pathway, to store excess solar PV electricity and low-price grid electricity for later use. The average price of electricity sold back to the grid in the *Hourly** and *Hourly*** pathways amounts to \$1,130/MWh and \$66/MWh, respectively, while the average price of electricity bought from the grid in these cases equals \$19/MWh and \$24/MWh, respectively. These price differences show how important the interaction between the grid and battery storage system can be for producing low-cost, electricity-based hydrogen. Again, these trends remain consistent for the Texas and New York production locations. Explore Section 9 of the SI for more detail on electricity-based hydrogen production operations.”

- Line 155: include the CO2 capture rates assumed in the three cases.

Completed.

Manuscript (Lines 217 - 219)

“Figure 4 also shows that the SMR-CCS (2) and ATR-CCS pathways, with a high percent carbon capture (96% and 95% capture, respectively), have a lower total LCOH than the SMR and SMR-CCS (1) pathways that have no or little carbon capture equipment installed (0% and 56% capture, respectively).”

- Line 196 and followings. This is a very nice finding!

Thank you.

- The electricity purchase price appear in figure 5 and not in the text. The upper limit of the sensitivity is 84 \$/MWh, which is not very high: Europe has seen peak prices above 700 €/MWh throughout 2022. Hopefully these will not repeat and will not be the annual average, but given that the work is doing a sensitivity assessment, the range could be enlarged. The same goes for natural gas price, which has now long remained above 50 €/MWh_{HHV} (peaked over 200 €/MWh_{HHV}).

These are good points. Bounds have been extended for natural gas prices between \$3/MMBTU and \$15/MMBTU (see Figure 5 in main text) to reflect possible futures in the United States. We could only vary electricity prices based on different price forecast data available to us from the NREL Cambium datasets.

Manuscript (Line 297)

- Line 222, the reason of varying between including and excluding PV upstream emissions should be described. Also, what is included in those upstream LCA should be clarified. Why is it not done for grid electricity, whose emission factors are assumed very low? (remember that a good NG-fired CCGT has CO₂ emissions in the order of 350-400 gCO₂/kWh just for the combustion at facility point).

The reasoning for including/excluding PV emissions has been described in the updated text. The reasoning is because it is not yet clear if solar PV emissions will be accounted for when figuring out what level of PTC a production pathway will achieve.

Manuscript (Lines 322 - 324)

“We vary treatment of GWP timeframe and embodied emissions in Figure 6 as it is still unclear whether life-cycle emissions of solar PV and other electricity-generating sources will be counted and which GWP timeframe will be used when determining the level of PTC achievable under 45V.”

- Line 226. Good point, agreed.

Thank you.

- Line 279 and following. This is true in the USA system with current schemes. But not in a system where

a "carbon tax" is applied on CO₂ emissions, such as the ETS scheme active in the EU. Might be interesting to deepen/mention.

We now mention the ETS scheme and how results may differ in the EU.

Manuscript (Lines 407 - 412)

"This makes the decision on which GHG emission accounting scheme to choose for 45V quite consequential to hydrogen developers looking to take advantage of U.S. tax incentives. It is worth noting, however, that jurisdictions like the European Union operate under an emissions trading system (ETS) instead of an incentive-based structure [17] which makes it more difficult for fossil-based hydrogen production to remain viable in these places."

- Line 305. Hydrogen compression is mentioned, which never appeared before. Why is that? Where is it located? It is not depicted in Figure 1. Is it accounted for in capital and operational costs? What is the considered pressure? ...

Updated in Figure 1a to show that hydrogen is compressed into hydrogen storage. Compression capital costs are included with storage costs and operating cost of compressors is accounted for on a kWh electricity per kg basis (see Table S.1 in Supporting Information). Manuscript has been updated in Introduction to mention hydrogen is stored at 200 bar.

Manuscript (Lines 62 - 65)

"All electricity-based production pathways explored in this study consider an onsite-solar photovoltaic (PV) facility with the option to include energy storage (battery or compressed hydrogen storage at 200 bar [20 MPa]) and/or a grid interconnection to increase delivery reliability."

- Line 309: 50% reduction of CO₂ emissions in a future grid where RES power plants are massive and energy storage are necessarily significant might not be in line with the LCA approach, which would require to include upstream emissions for those installations... Please verify the assumption and/or discuss it.

Upstream emissions from the renewables on the grid are considered in the updated emissions data we are using from Cambium.

Manuscript (Lines 438 - 444)

"GHG emissions associated with solar PV and grid electricity are accounted for on a life-cycle basis and are included in the electricity-based hydrogen production cost model. We assume a carbon intensity of 0.04 kg CO₂e per kWh_e for embodied solar PV emissions [18], [19] and use hourly carbon intensity data derived from NREL's 2035 Cambium Mid-case datasets to represent next decade emissions performance [14]. We modify these hourly emission datasets for our base case scenarios to include embodied emissions from electricity generating sources. Data and equations used to make this adjustment are available in Sections 1 and 3 of the SI."

- Line 365. A revenue of 56 \$/MWh is quite high (larger than expected LCOE) and may not be true in a

future high-RES electric system where the overgeneration of our plant will likely match with the overgeneration of many other plants. The analysis should include a case that consider zero remuneration of surplus generation (i.e., it is curtailed and not sold to grid).

We run sensitivities on the compensation rate for the PV/Storage/Grid* case and set it equal to zero in one case. When we set the compensation rate to zero, the LCOH increases by less than \$0.30/kg in Sacramento, CA and remains the same in the other two locations. Because compensation rate does not impact LCOH results significantly, we choose to leave it out of Figure 5. Compensation rate sensitivity results are available in Section 5 of the Supporting Information in Table S.6.

- Lines 376 and following. Please summarize reforming assumptions in a table (see also comment for techno-economic input data above).

Completed in updated version of manuscript. Please see Table 2 in the manuscript.

Manuscript (Lines 532 – 533)

“The primary input values used for each of the four fossil-based production pathways are shown in Table 2 with complete data available in Section 4 of the SI.”

Reviewer #2 (Remarks to the Author):

Author responses below are in **bolded** text, with updates to the manuscript reproduced in **blue bolded** text.

Leveling the Playing Field: A Cost Comparison of Various Hourly-Reliable, Net-Zero Hydrogen Production Pathways

Submitted to Nature Communications

In this manuscript, the authors investigate the future hydrogen costs with different production scenarios, namely electricity-based hydrogen with and without a grid, fossil-based hydrogen with different carbon capture methods. By comparing the cost components, the study aims to outline the development pathways for achieving net-zero hydrogen production. In addition, the impact of government incentive schemes has been investigated. Optimization models and simulation models have been used, with the support of data focusing on Sacramento and California region. The study highlights the importance of grid electricity in electricity-based hydrogen production. In addition, with US Inflation Reduction Act, the fossil-based production could be in favor, which could be against the initial motivation of the incentive scheme.

I have some major concerns, which need authors' consideration.

Major criticisms

1. There are quite many results of the study, often valid with the specific conditions of the study. But it is still unclear what are commonly known, what are the new findings in current study. A better presentation is needed to see the value of the study.

We believe the updated Discussion summarizes the new findings of this study well. We also addressed the first sentence of this comment by looking at more conditions (three different states) and discussing findings that were common no matter the region analyzed.

The three key takeaways of this study are:

- 1. Hydrogen delivery reliability is often overlooked. Without a grid connection, only tropical or equatorial regions can produce solar PV-based hydrogen without a significant overbuild and cost increase to achieve hourly reliability and net-zero emission targets.**

Manuscript (Lines 378 - 383)

“This study highlights that net-zero electricity-based hydrogen production with variable energy input becomes increasingly expensive with stricter reliability constraints and no grid connection; a point often overlooked in hydrogen production cost modeling efforts. Without an accessible grid connection, only tropical or equatorial countries with less seasonality in their solar resource may be able to supply hourly reliable hydrogen with less overbuild. In higher latitude locations, limited and seasonal solar resources would increase the relative attractiveness of fossil-based hydrogen production methods.”

2. In all locations analyzed, net-zero, hourly reliable, electricity-based hydrogen production can reach cost equivalence, if not cost savings, compared to fossil-based alternatives when natural gas leakage is considered by next decade.

Manuscript (Lines 387 - 392)

“In fact, the net-zero, hourly-reliable electricity-based hydrogen production pathways that include a grid connection can achieve cost-parity with, if not cost savings compared to, fossil-based alternatives by next decade (2030s). This assumes that the grid electricity purchased continues to decline in carbon intensity. Fossil-based production pathways are especially expensive when compared with grid-tied electricity-based solutions when considering both a cost for removing GHG emissions and high upstream natural gas leakage.”

3. When considering incentives in the United States, the hourly reliable electricity-based pathways explored can only achieve cost competitiveness with fossil-based alternatives if embodied emissions of electricity inputs are not counted in the hydrogen production emission accounting guidance.

Manuscript (Lines 403 - 409)

“With the 45Q and 45V tax credits included, fossil-based production pathways can remain cost-competitive with electricity-based pathways explored because of the high value of the 45Q tax credit, and the fact that the 45Q tax credit value is independent of the production facility emission intensity. Fossil-based pathways are especially favorable if embodied emissions of electricity inputs are counted in all production pathways. This makes the decision on which GHG emission accounting scheme to choose for 45V quite consequential to hydrogen developers trying to decide what type of production infrastructure to build out.”

2. One interesting aspect of the study is the pathway of hydrogen development, which also means that the evolving path of the cost components could be interesting. As the authors have the “Current, Next decade and Mid-century values” of input parameters, would it be possible to describe the development of the hydrogen costs?

We now include an expanded discussion around the current and mid-century cost results.

We felt it would be too much information to move current and mid-century results into the main text, but see the SI, Section 10, for these additional details.

Manuscript (Lines 189 - 191)

“See Figure S.23 of the SI to examine how the next-decade results shown in Figure 3 compare to those for current and mid-century timeframes. LCOH trends seen in Figure 3 remain consistent for current and mid-century results but with differing magnitudes due to the level of technology evolution.”

Manuscript (Lines 224 - 232)

“Figure S.24 of the SI also contains Figure 4 but is accompanied by cost comparisons using current and mid-century technology data. The current technology results in Figure S.24 show a similar trend to those in Figure 4 with the *PV/Storage/Grid** pathway being favorable over fossil-based pathways

when natural gas leakage is reaching 4%. It is also noticeable how uneconomic hourly-reliable hydrogen production costs are in the current timeframe when we set production to net-zero emissions. In contrast, the mid-century results show little additional cost for emissions removal since technology development itself will push hydrogen production pathways closer to net-zero. Note that there is inherent uncertainty in future energy system costs and so results in Figure S.24 should not be taken as forecasts, but as plausible scenarios defined using existing literature.”

3. I agree with the authors that grid electricity is important for hydrogen production, and also I believe that hydrogen production should be connected to grids, at least in the initial stage. But there are some issues which could be ignored by the authors.

- The electricity price in the grid should be stochastic, as long as it is traded in an open market. This is the case in many countries, and I assume the same in California. However, the model (Section 1 of the SI) seems only considering one price scenario (perhaps hourly price in one year). If this is true, would it be sufficient?

We have updated the analysis to include 3 different electricity marginal price forecasts from NREL’s Cambium dataset instead of using just one that was based off 2021 electricity prices in CA (CAISO). We find that using the NREL data leads to our model choosing more grid electricity purchases than before (see Figure 3 in main text) as the average grid pricing is lower in the NREL grid price cases than it was in the CAISO data we were using previously.

Manuscript (Lines 492 - 496)

“When modeling electricity-based pathways that include a grid interconnection, our next-decade cases use state-wide hourly marginal electricity pricing data from NREL’s Cambium database [14]. Our baseline results in Figures 3 and 4 use the 2035 Cambium Mid-case scenario electricity price data. For sensitivity analyses shown in Figures 4 and 5, we also include the Low Renewable Energy Cost and High Renewable Energy Cost scenarios from Cambium.”

Expanding to a stochastic optimization, based on probabilistic prices and imperfect information would be a valuable extension of this work, but falls outside the scope of this paper. We have added the following sentences to clarify this in the main text:

Manuscript (Lines 499 - 505)

“It is important to note that there is a high level of uncertainty in future marginal prices and demand charges across the United States and the next-decade electricity tariffs chosen in this study reflect only a few of many possible outcomes. We also do not account for how incremental electricity demand for hydrogen production facilities will themselves impact marginal electricity prices by next decade. Developing a stochastic optimization, based on probabilistic electricity prices and imperfect information, could be a valuable extension to this work, but is beyond the scope of this paper.”

- The electricity price may have a strong change in the future, especially if the electricity production sources are developed into renewables such as solar and wind. The intermitted supply eventually could make the grid price with a very high volatility (when marginal cost is used for pricing), which will then

have a significant impact on the grid-based hydrogen. As the study focuses on the hydrogen pathway, such impacts in the long-run should be relevant.

We address this by using hourly electricity price forecast data from NREL’s Cambium dataset for the year 2035. We find that the NREL data contains high grid price volatility that is expected as intermittent renewables increasingly dominate the grid mix. As an example, in the Mid-case scenario for 2035 in California, grid prices fluctuate from \$0-\$4,345/MWh over the course of the year.

- I am not sure what is the scale of electricity-based hydrogen production compared with the local electricity market. If it is sufficiently large, hydrogen production could affect electricity price, which will again affect the hydrogen price. This may not be an issue currently but could be a problem when hydrogen market is large.

We agree with this comment and now mention this phenomenon in the Methods section.

Manuscript (Lines 502 - 503)

“We also do not account for how incremental electricity demand for hydrogen production facilities will themselves impact marginal electricity prices by next decade.”

On lines 366-367, there are some related discussions. But the above concerns with electricity need to be more clearly elaborated.

We agree and have addressed this in the updated manuscript

4. The conclusion of the study is rather based on the specific location Sacramento and California. We then need to be more careful in interpreting the results. For example, the solar capacity factor is 31% (see Section 3 of the SI), this is much higher than the average value in other regions worldwide, which is typically 10%-20%. It might also be good to present what the major sources of electricity production in California are (current and future), in order to understand the grid carbon intensity values (lines 310-311).

We have updated the analysis to include results for California, New York, and Texas which have differing geographic and energy attributes. The solar PV capacity factors for each location are found in Table S.1 of the SI. We find that the high-level takeaways from this study remain consistent no matter what location we are analyzing in the United States. However, we found maintaining hourly hydrogen production reliability for off-grid electricity-based pathways is less expensive in lower latitude locations (El Paso, TX) since there is a favorable solar resource with low seasonal variation.

Manuscript (Lines 134 - 137)

“Noticeably, total LCOH for the *Hourly* case in Texas increases the least due to the facility being located at a lower latitude with less variation in solar PV output throughout the year. This finding would also apply to locations outside of the United States that are located at lower latitudes.”

5. I am not quite sure why a linear optimization approach can be used for defining the capacity of solar PV, battery, electrolyzer and hydrogen storage. If we see electricity supply from solar PV, it should be

random and follow a pattern of sine function. This means that the installed solar PV capacity and storage capacity should follow a non-linear relationship. Seasonal variation adds more complexity, and similar situation is applied to the hydrogen storage when the electrolyzer is restricted to the solar PV inputs. The authors also have similar concerns on lines 98-101, but the model is not reflecting this non-linear condition.

Linear optimization refers to the relationships between key variables in the modeled system, both in the objective function and in the constraints. As long as the cost function that the optimization is trying to minimize and the constraint equations that link key variables together (e.g. hydrogen production levels cannot exceed electrolyzer capacity) are all linear equations, based on additive and multiplicative relationships, the optimization remains linear. This is true even if the time series of input data takes a non-linear form, e.g. quasi-sinusoidal solar production levels.

We have added the following line to the Methods section to clarify this point:

Manuscript (Lines 446 - 448)

“Note that linear optimization refers to the structure of the equations within the optimization itself, and not the temporal profile of input data, such as solar electricity production, which is nonlinear over time.”

6. Conclusion: “A key finding from this study is that net-zero electricity-based hydrogen production with variable energy input becomes increasingly expensive with stricter reliability constraints and no grid connection.” This seems quite trivial to me for a solar PV and electrolyzer system, as the CF of solar PV has restricted the input to electrolyzer and therefor the capital costs are high in such a setting.

We have updated that part of the conclusion so that this somewhat self-evident statement it is no longer listed as a finding, but just a point that is often left out of hydrogen production cost reports. The key findings are fleshed out later in the conclusion.

Manuscript (Lines 378 - 380)

“This study highlights that net-zero electricity-based hydrogen production with variable energy input becomes increasingly expensive with stricter reliability constraints and no grid connection; a point often overlooked in hydrogen production cost modeling efforts.”

Other comments

7. Lines 105-109 (also figure 3): “...yearly reliable hydrogen delivery is lowest cost, with total LCOH more than doubling as reliability constraints increase to the hourly timeframe”. This is not surprising, as matching demand and supply is more difficult on the hourly basis, which will lead to overbuilt (over capacity of the electrolyzer and solar PV).

Agreed. We add more to this discussion by comparing results in the different states.

8. Lines 120 and 122: “...a grid connection allows for hourly delivery reliability at a 45% lower cost ...”

This is also not surprising, as the grid electricity now is a consistent feedstock to the electrolyzer, coupling the capacities of solar PV and electrolyzer is less important, and overbuilt is not necessary.

This sentence has been updated. We hypothesized mitigating grid emissions could cause the CO₂ mitigation cost to rise and potentially cause the LCOH to be higher than the PV/Storage case.

Manuscript (Lines 180 - 184)

“It is also important to highlight that there are costs incurred to connect to and use the grid system, as well as to remove grid-related emissions, in the *Hourly and *Hourly*** pathways that are not found in the off-grid pathway. However, the costs associated with using grid electricity are less than the savings created using that same electricity. This makes the grid connection beneficial for reducing the total LCOH, while also maintaining hourly delivery reliability.”**

9. Lines 307-308: “a carbon intensity of 0.04 kg CO₂e per kWh for upstream solar PV emissions ...” I am not sure of this. Since the upstream emission is mainly due to the equipment of Solar PV, the measure should base on per kW installed capacity, not on energy produced?

We use data from IPCC [19] and NREL [18] for this. The error bars in Figure 4 consider different values for the embodied emissions of solar PV since the actual value will depend on the location where the panels are installed.

10. Lines 340-341: “... the solar PV plant and the electrolyzer are \$500/kW (100MW system) and \$570/kW (1 MW system) ...” I understand the investment cost will include the scale factor. But still the above 100MW and 1MW systems seem not comparable. Some explanation?

We updated the numbers here so that both investment costs were scaled according to cost data for 100 MW systems. See Table 1 in the text.

Manuscript (Lines 471 - 472)

11. Section 1 in the SI: I interpret E_U as the electricity produced by solar PV and retained in the system (electrolyzer, compressor, battery), according to line 169. If this is correct, the constrain on line 177 should be revised: the grid electricity should not be reduced on the left of the equation, as it together with solar PV output will be the input for the electrolyzer.

Thank you for catching that. This has been updated. See Equation #2 in the SI.

12. Section 1 in the SI: What is the decision principle of curtailed electricity? I think when solar PV generates more electricity the input to be consumed by the electrolyzer system (including battery etc.), the rest is sent to the grid according to the balance constraints. But if the electricity grid price is high, can we send more electricity to the grid? Not quite clear about the decisions here.

Addressed in updated Methods section. The model will choose to send electricity back to the grid when marginal prices are high if it means LCOH savings overall.

Manuscript (Lines 496 - 498)

“Excess electricity generated from the solar PV, or pulled from the grid, can be stored in a battery and sold back to the grid during hours when these marginal prices are high if it means savings to the LCOH overall.”

13. Section 3 in the SI: I interpret Hydrogen Energy Content (33.3 kWh/kg) as electricity consumption per kg hydrogen production. Please check this value, I have the value which is about 49kWh/kg H₂.

By energy content we are referring to heating value, specifically the lower heating value of the hydrogen in this case. This has been clarified in Table S.1 of the SI.

To summarize, the study presents a very interesting topic and urgent issues. The study results could be very important for guiding the decision in energy transition, in particular for the regions considering hydrogen as a future alternative. However, data and analysis part need some further explanation and clarification to ensure the results. In addition, I believe that the presentation of results and conclusions can be improved, to indicate what are the main findings, as well as in which circumstances the findings are valid. I would therefore suggest a major revision of the manuscript.

We have addressed this comment by better clarifying and/or updating assumptions used in the model (i.e. electricity forecast price data, embodied emissions data, etc.) and by considering multiple locations for this study (California, New York, Texas).

We have also reworked the Discussion section to better highlight the key findings of this study. The 3 key findings are listed below:

- 1. Hydrogen delivery reliability is often overlooked. Without a grid connection, only tropical or equatorial regions can produce solar PV-based hydrogen without a significant overbuild and cost increase to achieve hourly reliability.**

Manuscript (Lines 378 - 383)

“This study highlights that net-zero electricity-based hydrogen production with variable energy input becomes increasingly expensive with stricter reliability constraints and no grid connection; a point often overlooked in hydrogen production cost modeling efforts. Without an accessible grid connection, only tropical or equatorial countries with less seasonality in their solar resource may be able to supply hourly reliable hydrogen with less overbuild. In higher latitude locations, limited and seasonal solar resources would increase the relative attractiveness of fossil-based hydrogen production methods.”

- 2. In all locations analyzed, net-zero, hourly reliable, electricity-based hydrogen production can reach cost equivalence, if not cost savings, compared to fossil-based alternatives when natural gas leakage is considered by next decade.**

Manuscript (Lines 387 - 392)

“In fact, the net-zero, hourly-reliable electricity-based hydrogen production pathways that include a grid connection can achieve cost-parity with, if not cost savings compared to, fossil-based alternatives by next decade (2030s). This assumes that the grid electricity purchased continues to decline in carbon intensity. Fossil-based production pathways are especially expensive when compared with grid-tied electricity-based solutions when considering both a cost for removing GHG emissions and high upstream natural gas leakage.”

3. When considering incentives in the United States, the hourly reliable electricity-based pathways explored can only achieve cost competitiveness with fossil-based alternatives if embodied emissions of electricity inputs are not counted in the hydrogen production emission accounting guidance.

Manuscript (Lines 403 - 409)

“With the 45Q and 45V tax credits included, fossil-based production pathways can remain cost-competitive with electricity-based pathways explored because of the high value of the 45Q tax credit, and the fact that the 45Q tax credit value is independent of the production facility emission intensity. Fossil-based pathways are especially favorable if embodied emissions of electricity inputs are counted in all production pathways. This makes the decision on which GHG emission accounting scheme to choose for 45V quite consequential to hydrogen developers trying to decide what type of production infrastructure to build out.”

Reviewer #3 (Remarks to the Author):

Author responses below are in **bolded** text, with updates to the manuscript reproduced in **blue bolded** text.

The manuscript addresses a topical problem in the energy sector around the hydrogen economy. I have a few comments.

As also highlighted in the manuscript, there have been numerous models introduced for investment decision analysis in various hydrogen vectors. The paper places its emphasis on the time resolution of past studies and argues that the granularity of the planning horizon (e.g. hourly profile) increases the decision accuracy. Indeed a valid point, but not an unknown fact. Consideration of any energy storage in planning requires a granular timeframe. However, if this is a limitation, there are other limitations as well that the manuscript doesn't address. For instance, seemingly the authors are using one capacity factor for solar output which significantly varies over locations. The use of solar+storage vs solar+storage+grid though is useful, doesn't add anything new to the current understanding.

The key contribution of this work is not the hourly resolution in our simulation, but the assessment of how much it would cost to produce a reliable supply of net-zero emission hydrogen from various pathways, including one that is constant from hour to hour.

We address location-dependent variation in capacity factor by including analysis for multiple locations to show how trends vary across various states with differing renewable resources. While much of this work seconds what is currently understood, the study is unique in the way production pathways are presented which could have led to results that break current understanding (e.g. including a CO₂ mitigation cost in the LCOH).

I have also concern about the results shown in Figure 3. The LCOH for reliability at hourly, daily, and monthly levels have relatively close values (say 5.5-6.5), but on a yearly basis, it suddenly drops to 2.5. This sharp decline from a monthly base to a yearly base is suspicious.

This drop is due to the requirement of seasonal storage for the monthly case which requires significant PV and storage overbuild (see Figure 3 in text). We also see an increase in embodied emissions from solar PV moving from the yearly to the monthly case because of the required solar PV overbuild. These emissions must be removed via the \$200/metric ton CO₂e removal cost which adds to the LCOH (see CO₂ mitigation cost in Figure 3).

The sharp increase from yearly to monthly reliability is also a function of location. In higher latitude locations, with more solar PV seasonal variability, higher amounts of storage are necessary. In locations closer to the equator, less seasonal variability in solar resource results in less storage and less of a solar PV overbuild (see Texas location in Figure 3). This last point is now noted in the manuscript (see below):

Manuscript (Lines 131 - 137)

“From the pathways with dedicated PV solar and no grid connection, *Yearly* reliable hydrogen delivery is lowest cost (\$2.88-\$3.88/kg H₂), with total LCOH increasing by 106%, 97%, and 35% compared to the *Hourly* case in California, New York, and Texas, respectively. Noticeably, total LCOH for the *Hourly* case in Texas increases the least due to the facility being located at a lower latitude with less variation in solar PV output throughout the year. This finding would also apply to locations outside of the United States that are located at lower latitudes.”

Last but not least, despite its generic title, this manuscript is limited to specific US policy scenarios.

While our paper focuses on the US, we address this feedback by mentioning the EU regulator market and how LCOH results could be changed.

Manuscript (Lines 409 - 412)

“It is worth noting, however, that jurisdictions like the European Union operate under an emissions trading system (ETS) instead of an incentive-based structure like the United States [17] which makes it difficult for fossil-based hydrogen production to remain viable in these places.”

We also mention how geographic location can impact LCOH results. These findings are valid in any location, not just the United States.

Manuscript (Lines 131 - 137)

“From the pathways with dedicated PV solar and no grid connection, *Yearly* reliable hydrogen delivery is lowest cost (\$2.88-\$3.88/kg H₂), with total LCOH increasing by 106%, 97%, and 35% compared to the *Hourly* case in California, New York, and Texas, respectively. Noticeably, total LCOH for the *Hourly* case in Texas increases the least due to the facility being located at a lower latitude with less variation in solar PV output throughout the year. This finding would also apply to locations outside of the United States that are located at lower latitudes.”

Manuscript (Lines 380 - 387)

“Without an accessible grid connection, only tropical or equatorial countries with less seasonality in their solar resource may be able to supply hourly reliable hydrogen with less overbuild. In higher latitude locations, limited and seasonal solar resources would increase the relative attractiveness of fossil-based hydrogen production methods. A grid-connected electricity-based system, however, enables increased flexibility in operations that mitigates the need for a system overbuild even in higher latitude locations, while maintaining the necessary hydrogen reliability that is required at refineries, ammonia plants, liquefaction plants, and other facilities.”

References:

- [1] J. Bracci, A. Brandt, S. M. Benson, G. Shrimali, and S. D. Saltzer, "Pathways To Carbon Neutrality in California: The Hydrogen Opportunity," Stanford, 2022. [Online]. Available: <https://sccc.stanford.edu/hydrogen-opportunity-report>.
- [2] International Energy Agency, "The Future of Hydrogen: Seizing Today's Opportunities," 2019. doi: 10.1787/1e0514c4-en.
- [3] S. Bruce *et al.*, "National Hydrogen Roadmap," 2018. [Online]. Available: https://www.csiro.au/-/media/Do-Business/Files/Futures/18-00314_EN_NationalHydrogenRoadmap_WEB_180823.pdf.
- [4] International Renewable Energy Agency, "Hydrogen: A Renewable Energy Perspective," Abu Dhabi, 2019. Accessed: Aug. 17, 2021. [Online]. Available: www.irena.org.
- [5] Lazard, "Levelized Cost of Hydrogen Analysis," 2021. Accessed: Oct. 05, 2022. [Online]. Available: <https://www.lazard.com/media/451779/lazards-levelized-cost-of-hydrogen-analysis-vf.pdf>.
- [6] T. Grube *et al.*, "A techno-economic perspective on solar-to-hydrogen concepts through 2025," *Sustain. Energy Fuels*, vol. 4, no. 11, pp. 5818–5834, Oct. 2020, doi: 10.1039/D0SE00896F.
- [7] L. Weimann, P. Gabrielli, A. Boldrini, G. J. Kramer, and M. Gazzani, "Optimal hydrogen production in a wind-dominated zero-emission energy system," *Adv. Appl. Energy*, vol. 3, p. 100032, Aug. 2021, doi: 10.1016/J.ADAPEN.2021.100032.
- [8] A. Al-Sharafi, A. Z. Sahin, T. Ayar, and B. S. Yilbas, "Techno-economic analysis and optimization of solar and wind energy systems for power generation and hydrogen production in Saudi Arabia," *Renew. Sustain. Energy Rev.*, vol. 69, pp. 33–49, Mar. 2017, doi: 10.1016/J.RSER.2016.11.157.
- [9] J. Gertner, "The Tiny Swiss Company That Thinks It Can Help Stop Climate Change," *The New York Times*, 2019.
- [10] S. Fuss *et al.*, "Negative emissions—Part 2: Costs, potentials and side effects," *Environ. Res. Lett.*, vol. 13, no. 6, p. 063002, May 2018, doi: 10.1088/1748-9326/AABF9F.
- [11] E. D. Sherwin, "Electrofuel Synthesis from Variable Renewable Electricity: An Optimization-Based Techno-Economic Analysis," *Environ. Sci. Technol.*, vol. 55, no. 11, pp. 7583–7594, Jun. 2021, doi: 10.1021/ACS.EST.0C07955/SUPPL_FILE/ES0C07955_SI_001.PDF.
- [12] J. A. Yarmuth, *H.R.5376 - Inflation Reduction Act of 2022*. Washington D.C., 2022.
- [13] Hydrogen Tools, "Hydrogen Production," 2016. <https://h2tools.org/hyarc/hydrogen-production> (accessed May 16, 2021).
- [14] P. Gagnon, B. Cowi, and M. Schwarz, "Cambium 2022 Data," 2023. [Online]. Available: <https://scenarioviewer.nrel.gov/>.
- [15] R. W. Howarth and M. Z. Jacobson, "How green is blue hydrogen?," *Energy Sci. Eng.*, vol. 9, no. 10, pp. 1676–1687, Oct. 2021, doi: 10.1002/ESE3.956.
- [16] C. Bauer *et al.*, "On the climate impacts of blue hydrogen production," *Sustain. Energy Fuels*, vol. 6, no. 1, pp. 66–75, Dec. 2021, doi: 10.1039/D1SE01508G.
- [17] "EU Emissions Trading System (EU ETS)," *European Commission*. https://climate.ec.europa.eu/eu-action/eu-emissions-trading-system-eu-ets_en (accessed Apr.

23, 2023).

- [18] National Renewable Energy Laboratory, "Energy Analysis: Life Cycle Assessment Harmonization," 2012. <https://www.nrel.gov/analysis/life-cycle-assessment.html> (accessed Sep. 29, 2022).
- [19] O. Edenhofer *et al.*, "Renewable energy sources and climate change mitigation: Special report of the intergovernmental panel on climate change," Cambridge University Press, 2012. doi: 10.1017/CBO9781139151153.

REVIEWER COMMENTS

Reviewer #1 (Remarks to the Author):

The revised manuscript shows a good improvement from the original submission, with the implementation of most changes and suggestions. Although some results are trivial (e.g., a grid-connected H₂ production system has better electrolysis CF), the numerical values are of interest. As final comments, I suggest:

- in the abstract, final sentence, the fact that USA is the observed region and origin of regulatory elements, should be stated/clarified
- in the discussion, some comments on generality and extension of observed results should be pointed out

Reviewer #2 (Remarks to the Author):

This is the revised version of manuscript with the original title "Leveling the Playing Field: A Cost Comparison of Various Hourly-Reliable, Net-Zero Hydrogen Production Pathways". I appreciate the authors' effort, which attempts to clarify the doubts in the comments. The manuscript has been improved. However, I still have some concerns which need the attention.

1. The authors summarized three key takeaways. About the first one, I do not agree with since the grid importance has been mentioned in existing studies. About the second one, I am not sure, by large, how important the factor of natural gas leakage is. About the third one concerning policies, it could be relevant. The previous comment "what are the new findings in current study" remains, and this is consistent with the opinion of Reviewer #3.
2. The study aims with hourly-reliable supply of hydrogen, which I interpreted as using electrolyzer at the full capacity, constantly (or with a converting factor of 70%, depending how this is explained). If I assume 500MW electrolyzer, and 10*500MW solar (10 times solar to "guarantee" the feed-in electricity to be hourly reliable), I end with investment cost of solar and electrolyzer = $(460+10*600)*1000*500=3.23*10^9$. With WACC=0.08, I have CRF = 0.0937 for 25 years, which will lead to capex = $(3.23*10^9*0.0937)/(250*1000*8760)=0.138\$/kg$, which is not fitting (less than) the numbers in Figure 3. In the solar-electrolyzer system, the capex should dominate the total cost, and this is often the reason that stand-alone (without grid) hydrogen system is more expensive. But the above numbers are not coming together, or I must be wrong somewhere.
3. The model has a resolution on hourly basis. I have raised the question before that "... electricity supply from solar PV, it should be random and follow a pattern of sine function ...". In the current manuscript and the modelling part of SI, it is still not clear if the supply of electricity from solar is on hourly basis, or with a lumping average (see table 1). In the figures of latter part SI, it seems that solar output is calculated on hourly basis, but again I cannot see if one year (or several years) data is used, or the data is based on selected weeks or months? In addition, where is the data source? This concern needs to be clarified.
4. In the study, why not present the capacity size of solar and storage, so that we have a better understanding of distribution of the costs.
5. About curtailed electricity, why the electricity can be sold back at the average price of 1130\$/MWh (line 174)? This is way too high.
6. The authors set the electrolyzer at the size of 500MW. There is an implicit assumption that electrolyzers are at a high scale level (to get a scale factor). Practically, with electrolyzer at 100MW scale level, the coupling solar system needs to be several times of that level, and thereby the installation needs substantial space. This could be a practical problem.

I would suggest some further revision and clarification of the manuscript.

Reviewer #1:

Author responses below are in **bolded** text, with updates to the manuscript reproduced in **blue bolded** text.

The revised manuscript shows a good improvement from the original submission, with the implementation of most changes and suggestions. Although some results are trivial (e.g., a grid-connected H2 production system has better electrolysis CF), the numerical values are of interest.

We agree with this assessment and further point this out in the discussion.

Manuscript [Lines 395 – 399]:

“Including a net-zero emission constraint to the LCOH is novel, placing climate solutions with different emissions profiles on a level techno-economic playing field. Therefore, the large differences in LCOH between the *Yearly* and *Hourly* reliability scenarios, and between the *Hourly* and *Hourly scenarios presented in Figure 3 are noteworthy.”**

As final comments, I suggest:

- in the abstract, final sentence, the fact that USA is the observed region and origin of regulatory elements, should be stated/clarified

We have clarified in the Abstract that results are specific to the United States but can be applied to other regions with a similar climate and grid mix when not considering U.S. policy. Thank you for the suggestion.

Manuscript [Lines 16 – 23]:

“The optimization-based techno-economic model developed for electrolysis production pathways suggests solutions in the United States (California, Texas, and New York locations) that are grid-tied can be lower cost (\$2.02-\$2.88/kg) than fossil-based pathways with upstream natural gas leakage greater than 4% (\$2.73-\$5.94/kg). These results also apply to regions outside of the U.S. with a similar climate and electric grid mix to the locations used in this study. Focusing on a next-decade scenario under the United States regulatory environment, where U.S. Inflation Reduction Act tax credits are included and net-zero emission constraints are omitted, results...”

- in the discussion, some comments on generality and extension of observed results should be pointed out

We have addressed this comment by including the following lines in the discussion section about possible future work and how our results would compare if performed using policy schemes and energy attributes from other regions of the world.

Manuscript [Lines 410 – 413]

“In a future study, it would be worth examining locations outside of the United States with higher grid carbon intensities and relaxed emission reduction goals to see how hourly reliable, net-zero

electricity-based pathways with a grid connection fare against fossil-based and off-grid renewable electrolysis alternatives.”

Manuscript [Lines 437 – 439]

“Analyzing various hydrogen production pathway costs using policy schemes outside of the United States, such as ETS in the European Union, would be a valuable extension to this work to show what pathways are preferable in different regions of the world.”

Reviewer #2 (Remarks to the Author):

Author responses below are in **bolded** text, with updates to the manuscript reproduced in **blue bolded** text.

This is the revised version of manuscript with the original title “Leveling the Playing Field: A Cost Comparison of Various Hourly-Reliable, Net-Zero Hydrogen Production Pathways”. I appreciate the authors’ effort, which attempts to clarify the doubts in the comments. The manuscript has been improved. However, I still have some concerns which need the attention.

Thank you for the comment. We have addressed each of these concerns by either updating relevant sections of the manuscript or by providing further reasoning for leaving relevant sections of the manuscript unchanged.

1. The authors summarized three key takeaways. About the first one, I do not agree with since the grid importance has been mentioned in existing studies. About the second one, I am not sure, by large, how important the factor of natural gas leakage is. About the third one concerning policies, it could be relevant. The previous comment “what are the new findings in current study” remains, and this is consistent with the opinion of Reviewer #3.

On the first takeaway, we agree that the importance of a grid connection has been mentioned in previous studies. The key point is that our results are for “net-zero” production pathways such that each pathway explored includes a cost for mitigating all residual CO₂ emissions. This means that results from our study may not exactly align with previous studies. For example, if we were in a location with a high grid carbon intensity, grid-connected electrolysis could be less favorable than an off-grid renewable electrolysis pathway with hourly reliability targets. We have added a few sentences to the discussion to make these points.

Manuscript [Lines 394 – 399]

“This study highlights that net-zero electricity-based hydrogen production with variable energy input becomes increasingly expensive with stricter reliability constraints and no grid connection. Including a net-zero emission constraint to the LCOH is novel, placing climate solutions with different emissions profiles on a level techno-economic playing field. Therefore, the large differences in LCOH between the *Yearly* and *Hourly* reliability scenarios, and between the *Hourly* and *Hourly scenarios presented in Figure 3 are noteworthy.”**

Manuscript [Lines 408 – 413]

“This assumes that the grid electricity purchased continues to decline in carbon intensity such that the CO₂ removal cost for mitigating electric grid-related emissions is marginal. In a future study, it would be worth examining locations outside of the United States with higher grid carbon intensities and relaxed emission reduction goals to see how hourly reliable, net-zero electricity-based pathways with a grid connection fair against fossil-based and off-grid renewable electrolysis alternatives.”

On the second takeaway, we now mention by how much natural gas leakage can potentially add to the cost of producing net-zero fossil-based hydrogen based on the results in Figure 4.

Manuscript [Lines 414 – 417]

“In this study we also find that fossil-based production pathways are especially expensive when compared with grid-tied electricity-based solutions in the United States when considering both a cost for removing residual GHG emissions and high upstream natural gas leakage. With these assumptions in place, fossil-based production costs increase over two-fold (see Figure 4).”

On your comment about “what are the new findings in current study”, we agree some of the findings of this study may appear trivial, but the method used to reach these findings was new. Our net-zero emission constraint is reflective of national policies and corporate commitments across the globe. This means our findings are, in many cases, more decision-relevant than previous studies, which did not have as elegant a method of joining techno-economic analysis and life-cycle assessment. Please see below for the section of the manuscript where we mention these points.

Manuscript [Lines 42 – 57]

“First, we include a CO₂ removal cost that ensures both fossil-based and electricity-based hydrogen production pathways have net-zero life-cycle emissions, thus making a fair comparison between different technologies. This avoids a situation where hydrogen production pathways with different GHG intensities are compared on a simple cost per unit of hydrogen, without accounting for the differences in associated emissions. The zero-emission constraint is also reflective of national policies and corporate commitments made around the world, therefore making our findings decision-relevant. Second, our model accounts for the consistency of hydrogen production as a function of time. This is a major factor for electrolytic hydrogen pathways, which generally aim to leverage intermittent low-cost renewable electricity resulting in a hydrogen production profile that can fluctuate on an hourly, daily, and seasonal basis. Using such sources could be a challenge for end consumers requiring consistent delivery. In many prior studies, reliability distinctions are either unclear or not addressed [1]–[5], and studies that do focus on hourly-resolved power-to-hydrogen systems tend to not have as elegant a method of joining techno-economic analysis and life-cycle assessment [6]–[8]. By explicitly addressing these two inconsistencies in prior comparisons, this work will provide clarity to academics, policy makers, and companies on the cost of producing reliable, zero-carbon hydrogen and can help to guide investment in hydrogen infrastructure.”

2. The study aims with hourly-reliable supply of hydrogen, which I interpreted as using electrolyzer at the full capacity, constantly (or with a converting factor of 70%, depending how this is explained). If I assume 500MW electrolyzer, and 10*500MW solar (10 times solar to “guarantee” the feed-in electricity to be hourly reliable), I end with investment cost of solar and electrolyzer = $(460+10*600)*1000*500=3.23*10^9$. With WACC=0.08, I have CRF = 0.0937 for 25 years, which will lead to capex = $(3.23*10^9*0.0937)/(250*1000*8760)=0.138\$/\text{kg}$, which is not fitting (less than) the numbers in Figure 3. In the solar–electrolyzer system, the capex should dominate the total cost, and this is often the reason that stand-alone (without grid) hydrogen system is more expensive. But the above numbers are not coming together, or I must be wrong somewhere.

Thank you for the comment. An hourly-reliable supply of hydrogen is achieved by using a combination of solar PV, energy storage, and in some cases an electric grid connection. In a case with only energy storage and solar PV, the system does not need to build the solar PV at 10x, rather it should build it at

about 4x if the solar capacity factor is around 25%. Energy storage's role in enabling hourly-reliable hydrogen is mentioned in the manuscript, lines 66-69, and illustrated in Figure 1.

Manuscript [Lines 68 – 71]

“All electricity-based production pathways explored in this study consider an onsite-solar photovoltaic (PV) facility with the option to include energy storage (battery or compressed hydrogen storage at 200 bar [20 MPa]) and/or a grid interconnection to increase delivery reliability.”

Also, we found a slight error in your calculations which made your result too low. We have corrected it in **red** below:

$$\text{capex} = (3.23 \times 10^9 \times 0.0937) / (250 \times 1000 \times 365) = 3.317 \$/\text{kg}$$

We have also added Table 1 to the manuscript [Line 190], containing system component sizes for each electricity-based pathway, to aid in understanding the distribution of costs.

3. The model has a resolution on hourly basis. I have raised the question before that “... electricity supply from solar PV, it should be random and follow a pattern of sine function ...”. In the current manuscript and the modelling part of SI, it is still not clear if the supply of electricity from solar is on hourly basis, or with a lumping average (see table 1). In the figures of latter part SI, it seems that solar output is calculated on hourly basis, but again I cannot see if one year (or several years) data is used, or the data is based on selected weeks or months? In addition, where is the data source? This concern needs to be clarified.

We have addressed this comment by mentioning that the model is hourly resolved using grid electricity price data and system costs relevant in the year 2035. Therefore, we are only using one year of data with 8760 timestamps for each hour of a year. We also specify that we are using hourly solar capacity factor data from NREL's System Advisor Model. Hourly electricity price and solar capacity factor data used is available in the Source Data spreadsheet accompanying the manuscript.

Manuscript [Lines 67 – 68]

“The model is hourly resolved with a total of 8760 timesteps representing each hour of a specified year.”

Manuscript [Lines 71 – 73]

“Hourly grid electricity prices are derived from the National Renewable Energy Laboratory's (NREL) Mid-Case scenario in the Cambium database [9] and hourly solar capacity factor data is derived from NREL's System Advisor Model [10].”

Manuscript [Lines 267 - 270]

“To capture uncertainty and variability in hourly grid electricity prices, we use the National Renewable Energy Laboratory's (NREL's) 2035 Cambium Mid-Case as our central scenario and the 2035 Low Renewable Energy Cost and High Renewable Energy Cost scenarios as the low and high price bounds, respectively [9].”

4. In the study, why not present the capacity size of solar and storage, so that we have a better understanding of distribution of the costs.

We have added Table 1 to the main text with system component sizes for the Sacramento, California location. System component sizes for the other locations are shown in Section 9 of the SI. Thank you for the suggestion.

Manuscript [Line 190]

5. About curtailed electricity, why the electricity can be sold back at the average price of 1130\$/MWh (line 174)? This is way too high.

The hourly electricity price data we are using from NREL's Cambium database consists of some hours with electricity prices above \$1130/MWh in California for the year 2035. The optimization model decides to send electricity back to the grid when the prices are high which leads to the surprising result. See below the electricity price data we use from the Cambium Mid Case Scenario for California in 2035. This data is also available in the Source Data file accompanying the manuscript.

6. The authors set the electrolyzer at the size of 500MW. There is an implicit assumption that electrolyzers are at a high scale level (to get a scale factor). Practically, with electrolyzer at 100MW scale level, the coupling solar system needs to be several times of that level, and thereby the installation needs substantial space. This could be a practical problem.

We agree that there could be issues with space requirements for renewable electrolysis facilities of 500 MW or larger. We now include a note acknowledging space constraints. This can be found in the Methods section when describing the optimization model.

Manuscript [Lines 463 – 465]

"In the current form of the model, we do not consider space constraints for large system components, such as solar PV or hydrogen storage, but acknowledge this could also impact optimal facility size, production method, and location for a new-build hydrogen production plant."

I would suggest some further revision and clarification of the manuscript.

We hope to have addressed each of your suggestions. Thank you very much for your help in strengthening the quality of this manuscript. We are appreciative of your time and thoughtfulness.

References

- [1] J. Bracci, A. Brandt, S. M. Benson, G. Shrimali, and S. D. Saltzer, "Pathways To Carbon Neutrality in California: The Hydrogen Opportunity," Stanford, 2022. [Online]. Available: <https://sccc.stanford.edu/hydrogen-opportunity-report>.
- [2] International Energy Agency, "The Future of Hydrogen: Seizing Today's Opportunities," 2019. doi: 10.1787/1e0514c4-en.
- [3] S. Bruce *et al.*, "National Hydrogen Roadmap," 2018. [Online]. Available: https://www.csiro.au/-/media/Do-Business/Files/Futures/18-00314_EN_NationalHydrogenRoadmap_WEB_180823.pdf.
- [4] International Renewable Energy Agency, "Hydrogen: A Renewable Energy Perspective," Abu Dhabi, 2019. Accessed: Aug. 17, 2021. [Online]. Available: www.irena.org.
- [5] Lazard, "Levelized Cost of Hydrogen Analysis," 2021. Accessed: Oct. 05, 2022. [Online]. Available: <https://www.lazard.com/media/451779/lazards-levelized-cost-of-hydrogen-analysis-vf.pdf>.
- [6] T. Grube *et al.*, "A techno-economic perspective on solar-to-hydrogen concepts through 2025," *Sustain. Energy Fuels*, vol. 4, no. 11, pp. 5818–5834, Oct. 2020, doi: 10.1039/D0SE00896F.
- [7] L. Weimann, P. Gabrielli, A. Boldrini, G. J. Kramer, and M. Gazzani, "Optimal hydrogen production in a wind-dominated zero-emission energy system," *Adv. Appl. Energy*, vol. 3, p. 100032, Aug. 2021, doi: 10.1016/J.ADAPEN.2021.100032.
- [8] A. Al-Sharafi, A. Z. Sahin, T. Ayar, and B. S. Yilbas, "Techno-economic analysis and optimization of solar and wind energy systems for power generation and hydrogen production in Saudi Arabia," *Renew. Sustain. Energy Rev.*, vol. 69, pp. 33–49, Mar. 2017, doi: 10.1016/J.RSER.2016.11.157.
- [9] P. Gagnon, B. Cowiestoll, and M. Schwarz, "Cambium 2022 Data," 2023. [Online]. Available: <https://scenarioviewer.nrel.gov/>.
- [10] National Renewable Energy Laboratory, "System Advisor Model," 2020. <https://sam.nrel.gov/> (accessed Sep. 29, 2022).

REVIEWERS' COMMENTS

Reviewer #2 (Remarks to the Author):

I thank the authors for the clarification about my comments. I am in general satisfied with the revision, as the model and its presentation have been improved, and the study results become more convincing. I would suggest accepting the manuscript.

Reviewer #2 (Remarks to the Author):

Author responses below are in **bolded** text.

I thank the authors for the clarification about my comments. I am in general satisfied with the revision, as the model and its presentation have been improved, and the study results become more convincing. I would suggest accepting the manuscript.

We thank you again for your feedback. It has helped to improve the quality of this paper greatly.